# Communication-Efficient Language Model Training Scales Reliably and Robustly: Scaling Laws for DiLoCo

**Zachary Charles**
Google Research
zachcharles@google.com

**Gabriel Teston**
Google Search
teston@google.com

**Lucio Dery**
Google DeepMind
ldery@google.com

**Keith Rush**
Google DeepMind
krush@google.com

**Nova Fallen**
Google Research
nfallen@google.com

**Zachary Garrett**
Google Research
zachgarrett@google.com

**Arthur Szlam**
Google DeepMind
aszlam@google.com

**Arthur Douillard**
Google DeepMind
douillard@google.com

## Abstract

As we scale to more massive machine learning models, the frequent synchronization demands inherent in data-parallel approaches create significant slowdowns, posing a critical challenge to further scaling. Recent work [11, 24] develops and analyzes an approach (DiLoCo) that relaxes synchronization demands via periodic synchronization. However, these works do not carefully analyze how DiLoCo's behavior changes with model size. In this work, we study the scaling law behavior of DiLoCo when training LLMs under a fixed compute budget. We focus on how algorithmic factors, including number of model replicas, hyperparameters, and token budget affect training in ways that can be accurately predicted via scaling laws. We find that DiLoCo scales both predictably and robustly with model size. When well-tuned, DiLoCo scales better than data-parallel training with model size, and can outperform data-parallel training even at small model sizes. Our results showcase a more general set of benefits of DiLoCo than previously documented, including increased optimal batch sizes, improved downstream generalization with scale, and improved evaluation loss for a fixed token budget.

## 1 Introduction

Large language models (LLMs) are typically trained via large-batch distributed data-parallel methods. However, bandwidth and communication constraints can become bottlenecks at larger scales due to frequent synchronizations, posing a critical challenge to further scaling. As a remedy, Douillard et al. [11] propose DiLoCo (*Distributed Low-Communication*), a generalization of algorithms like Local SGD [34, 54] and FedAvg [36], which enables training of LLMs in parallel across "islands" of compute (such as datacenters connected via low-bandwidth networks) by performing parallel training of models with only periodic synchronization. While empirically successful, DiLoCo exists within a broader context of work on communication-efficient LLM training methods [58, 43, 56, 3, 24, 30]. Due to the breadth of related work, we defer a more thorough overview to Section A.

Unlike communication-reduction methods such as quantization and sparsification, DiLoCo fundamentally alters training dynamics [49]. While Douillard et al. [11] and Jaghouar et al. [24] show

39th Conference on Neural Information Processing Systems (NeurIPS 2025).

that DiLoCo yields comparable evaluation metrics to data-parallel training at moderate model scales, it is unclear how data-parallel training and DiLoCo compare at larger model scales. Moreover, DiLoCo has extra hyperparameters not present in data-parallel training that may be computationally prohibitive to tune at large enough scales. This points to the need for DiLoCo *scaling laws*. We focus on two specific scaling laws: (1) predictions for evaluation loss as a function of model size and (2) predictions for optimal hyperparameter choices for a given model size (which can obviate the need to perform expensive hyperparameter tuning). In both cases, we are explicitly interested in how these compare to analogous scaling laws for data-parallel training.

Throughout we consider the task of pre-training a model of size $N$ on $D$ tokens. We are concerned with predicting, for both data-parallel training and DiLoCo training, and as a function of $N$: (1) the evaluation loss $L$ after training, computed on a held-out set, and (2) optimal hyperparameter settings. One path towards scaling laws for DiLoCo would be to view them as modifications of scaling laws for data-parallel training [26, 20]. However, the facets of DiLoCo that are key to its communication-efficiency also make such an approach infeasible. First, DiLoCo operates by training $M$ models in parallel, with periodic synchronization every $H$ steps. The values of $M$ and $H$ depend on the ecosystem of compute available (such as bandwidth across datacenters), and are absent in scaling laws for data-parallel training. Second, DiLoCo uses a bi-level optimization framework; each model replica performs data-parallel training, but upon synchronization we apply an "outer" optimization step [11]. This means that DiLoCo has "outer" hyperparameters not present in Data-Parallel training that cannot be inferred from data-parallel hyperparameter scaling laws.

**Contributions.** We develop scaling laws for data-parallel training and DiLoCo from the ground up. Fixing the number of tokens $D$ to be the "Chinchilla-optimal" number of tokens [20], we model the evaluation loss and optimal hyperparameters as functions of model size $N$ and (for DiLoCo) the number of replicas $M$. [1] We empirically estimate these functions using the final evaluation loss attained by models trained with both algorithms for varying hyperparameters (including learning rate, batch size, and "outer" learning rate for DiLoCo), model sizes (varying $N$ over 9 model sizes ranging from 35 million to 2.4 billion parameters), and numbers of DiLoCo replicas $M$.

Our scaling laws predict that in many settings, the more communication-efficient DiLoCo algorithm actually yields better evaluation loss than data-parallel training for the same token budget. Utilizing our scaling laws to predict the hyperparameters for DiLoCo, we tested these predictions when training models with 4 billion and 10 billion parameters. The scaling laws proved accurate, with DiLoCo with $M = 2$ replicas outperforming data-parallel training as predicted, even while reducing total communication by a factor of over 100.

We show that DiLoCo incurs a variety of benefits in comparison to data-parallel training, including (1) increased optimal batch size, allowing for greater horizontal scalability, (2) greater reductions in evaluation loss as model size increases, and (3) significantly less wall-clock training time. One potentially surprising finding: DiLoCo improves training even when communication is not a bottleneck. DiLoCo with $M = 1$ (an enhanced version of the Lookahead optimizer [62]) does not reduce communication but achieves lower evaluation loss at all model scales. It is also more robust to larger batch sizes, greatly reducing wall-clock training time.

## 2   Preliminaries

Table 1: General Notation

| Symbol | Meaning |
| --- | --- |
| $\theta$ | Model weights |
| $N$ | Model size |
| $L$ | Evaluation loss |
| $T$ | Training steps |
| $D$ | Token budget |
| $C$ | Total FLOPs |

Table 2: Algorithm-Specific Notation

| Symbol | Data-Parallel | DiLoCo |
| --- | --- | --- |
| $\gamma$ | Learning rate | Inner learning rate |
| $\eta$ | – | Outer learning rate |
| $B$ | Batch size | Global batch size |
| $M$ | – | DiLoCo replicas |
| $H$ | – | Synchronization cadence |

---

[1]While we fix $H = 30$ for these scaling laws, we provide extensive ablations on the role of $H$ in Section 5.

Throughout, we let $\theta$ denote the model parameters. We let $\theta^{(t)}$ denote the model at step $t$. Since DiLoCo operates on $M$ parallel model parameters, we will use subscript notation $\theta_m$ to denote the $m$-th model. When there is no subscript, the parameters are assumed to be replicated across all DiLoCo replicas. For a batch of data $x$, we let $f(\theta, x)$ denote the loss of $\theta$ on the batch of data.

DiLoCo [11] is a technique designed for training models in the presence of communication constraints. It is motivated by the training of large models across devices that are not all connected by low-latency bandwidth. To avoid incurring latency costs, DiLoCo trains $M$ models in parallel (ideally, training each one with co-located compute connected via low latency bandwidth), only synchronizing the models every $H$ steps. This is similar to the FedOpt algorithm used in federated learning [47], but with the important difference that the replicas maintain their inner optimizer state across rounds.

DiLoCo applies a bi-level optimization framework across multiple models: each DiLoCo replica has its own model $\theta_m^{(t)}$, and there is a global model $\theta^{(t)}$. At every step, each replica takes an *inner* optimization step (InnerOpt). Every $H$ steps, each replica computes the $\Delta_m^{(t)} = \theta^{(t-H)} - \theta_m^{(t)}$, the difference between the replica's current model and the most recent global model. We average these differences across replicas, resulting in $\Delta^{(t)}$ which we refer to as an *outer gradient*. We treat this as a gradient estimate of the outer model[2] and an outer optimization step (OuterOpt) to the outer model $\theta^{(t-H)}$. This yields an updated outer model $\theta^{(t)}$ which is broadcast to all replicas and set as their current inner model. We give full pseudo-code for DiLoCo in Algorithm 1.

Throughout, we perform model training via distributed data-parallel training (Data-Parallel), and DiLoCo. In Data-Parallel, at each step we distribute a batch of $B$ tokens across workers. We then compute a batch gradient and perform optimization with a learning rate of $\gamma$. In DiLoCo, at each step $t$, we take a global batch of tokens of size $B$ consisting of $B/S$ sequences each of length $S$. We partition the $B/S$ sequences across the $M$ DiLoCo replicas, so each replica receives $B/SM$ sequences, each of length $S$. Thus, the *global* token batch size is $B$, but each DiLoCo replica uses a local token batch size $B/M$. As in Data-Parallel, each replica computes a batch gradient and applies an inner optimization step with learning rate of $\gamma$. Unlike Data-Parallel, DiLoCo does outer optimization (on outer-gradients computed in parameter space) every $H$ steps with learning rate $\eta$.

When comparing Data-Parallel and DiLoCo, we use the same model size $N$ and token budget $D$. When computing evaluation loss $L$ on some held-out set, for Data-Parallel we use the current

---

**Algorithm 1** DiLoCo

**Require:** Loss function $f(\theta, x)$, batch size $B$, number of replicas $M$, synchronization cadence $H$, initial model weights $\theta^{(0)}$, data shards $\{\mathcal{D}_1, \ldots, \mathcal{D}_M\}$
**Require:** Optimizers InnerOpt and OuterOpt
1: $\forall m, \theta_m^{(0)} \leftarrow \theta^{(0)}$
2: **for** step $t = 1 \ldots T$ **do**
3:     **parallel for** replica $m = 1 \ldots M$ **do**
4:         Receive a batch $x_m^{(t)} \sim \mathcal{D}_m$ of size $B/M$
5:         $g_m^{(t)} \leftarrow \nabla_\theta f(\theta_m^{(t-1)}, x_m^{(t)})$
6:         $\theta_m^{(t)} \leftarrow \text{InnerOpt}(\theta_m^{(t-1)}, g_m^{(t)})$
7:     **end parallel for**

8:     **if** $t \bmod H = 0$ **then**
9:         $\Delta_m^{(t)} \leftarrow \theta^{(t-H)} - \theta_m^{(t)}$
10:        $\Delta^{(t)} \leftarrow \frac{1}{M} \sum_{m=1}^{M} \Delta_m^{(t)}$
11:        $\theta^{(t)} \leftarrow \text{OuterOpt}(\theta_m^{(t-H)}, \Delta^{(t)})$
12:        $\forall m, \theta_m^{(t)} \leftarrow \theta^{(t)}$

---

model, and for DiLoCo we use the most recent global model. We summarize algorithm-independent notation in Table 1 and algorithm-specific notation in Table 2. An important comparison is Data-Parallel versus DiLoCo with $M = 1$. While similar, they are not identical as DiLoCo with $M = 1$ uses an outer optimizer step with optimizer OuterOpt, which is often set to SGD with Nesterov momentum [11].[3]

## 3 Experimental Methodology

**Model architecture.** We use a Chinchilla-style decoder-only transformer [20]. As suggested by Wortsman et al. [60] and Jaghouar et al. [24], we use QK-LayerNorm to reduce sensitivity to learning

---

[2]Note that $\Delta^{(t)}$ is generally not a gradient of any function, as it can evince non-conservative dynamics [5].
[3]This is similar in spirit to the fast- and slow-momentum steps in AdEMAMix [41], but yields different training dynamics since it uses a gradient estimate computed by linearizing across multiple training steps.

Table 3: Model details, including size, number of layers, layer dimensions, and token budgets. For the larger models (4B and 10B) we use scaling laws to predict optimal hyperparameters, rather than performing extensive hyperparameter tuning.

| Model Scale | Transformer Layers | Attention Heads | QKV Dimension | Hidden Dimension | Token Budget | Hyperparameter Sweep |
|---|---|---|---|---|---|---|
| 35M | 6 | 8 | 512 | 2,048 | 70M | ✓ |
| 90M | 9 | 12 | 768 | 3,072 | 1.8B | ✓ |
| 180M | 12 | 16 | 1,024 | 4,096 | 3.6B | ✓ |
| 330M | 15 | 20 | 1,280 | 5,120 | 6.6B | ✓ |
| 550M | 18 | 24 | 1,536 | 6,144 | 11B | ✓ |
| 1.3B | 24 | 32 | 2,048 | 8,192 | 26B | ✓ |
| 2.4B | 30 | 40 | 2,560 | 10,240 | 48B | ✓ |
| 4B | 36 | 48 | 3,072 | 12,288 | 80B | ✗ |
| 10B | 48 | 64 | 4,096 | 16,384 | 200B | ✗ |

rate. We also use z-loss regularization [8]. We use a vocabulary size of 32,768: 32,000 in-vocabulary words, and extra tokens for BOS and out-of-vocabulary. We pack multiple sequences into each batch, with a sequence length of 2,048 throughout. We pre-train a family of models, varying the number of transformer layers, number of attention heads, QKV dimension, and feed-forward layer hidden dimension (see Table 3). We use the Chinchilla-optimal token budget [20] unless otherwise noted. We do extensive hyperparameter sweeps on all models except the two largest (4B and 10B).

**Datasets.** Unless otherwise noted, we use the train split of the C4 dataset for training [45]. We report evaluation metrics on C4's held-out validation set. We compute downstream zero-shot evaluation metrics on 3 tasks: HellaSwag [61], Piqa [4], and Arc-Easy [9]. In overtraining ablations (Section 5), we use the Dolma dataset [53] instead of C4, to avoid doing more than a single epoch of training.

**Optimizers.** We use AdamW [32] as the optimizer for Data-Parallel and inner optimizer for DiLoCo, with $\beta_1 = 0.9$, $\beta_2 = 0.99$. Following [59], we use a weight decay parameter of $\lambda = T^{-1}$ where $T$ is the number of training *steps*. We do 1000 steps of warmup followed by cosine learning rate decay to 5% of the peak learning rate. We clip (inner) gradients to a norm of 1. We do not clip outer gradients. For DiLoCo, we use SGD with Nesterov momentum [55] as the outer optimizer, using a momentum of 0.9 and constant outer learning rate. Unless otherwise specified, we set $H = 30$.

**Implementation.** We use a modified version of NanoDO [31] that uses DrJAX [49] to parallelize training steps across replicas. We use bfloat16 representation of model weights and gradients.

**Idealized wall-clock time.** For each experiment, we compute an idealized end-to-end wall-clock time for training. Our model assumes that we are training a model across multiple datacenters. Within a datacenter, we have a high-bandwidth network. Across datacenters, we have a high-, medium-, or low-bandwidth network. For details on the idealized wall-clock time, see Section B.

**Scaling law experiments.** We perform comprehensive hyperparameter sweeps for Data-Parallel and DiLoCo on models ranging from 35M to 2.4B We sweep over the learning rate $\gamma$ using integer powers of $\sqrt{2}$ and batch size $B$ using powers of 2. For DiLoCo, we train using $M = 1, 2, 4, 8$ and sweep the outer learning rate $\eta$ over $\{0.2, 0.4, 0.6, 0.8, 1.0\}$. We sweep (inner) learning rate and batch size as needed until the minimum loss value is obtained on an interior point. Using this data, we derive scaling laws to predict evaluation loss and optimal hyperparameters for larger models. We use the predicted hyperparameters to train models with 4B and 10B parameters, in order to validate the scaling laws empirically.

## 4 Empirical Findings

Before discussing our process for fitting the specific scaling laws, we talk about the empirical results and four critical findings that are worth highlighting independently.

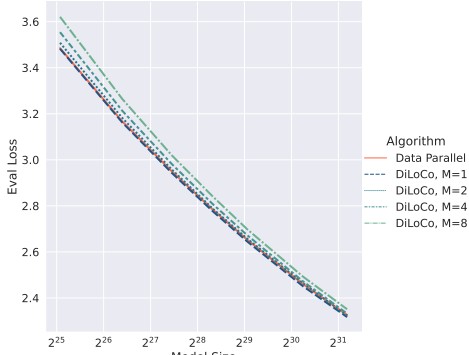

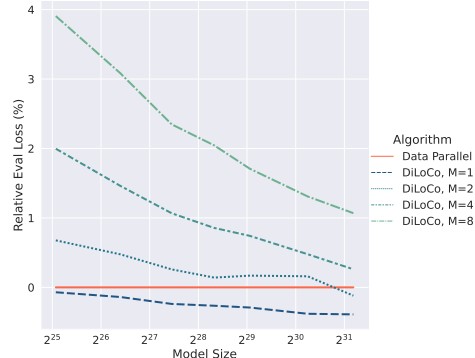

(a) Evaluation loss for various algorithms, as a function of model size $N$.

(b) Percentage difference in evaluation loss, relative to Data-Parallel.

Figure 1: We compare Data-Parallel to DiLoCo for varying model sizes $N$. For all $M$, DiLoCo improves monotonically with respect to Data-Parallel as $N$ increases.

**Finding 1: DiLoCo's evaluation loss improves relative to Data-Parallel as $N$ increases.** We give the evaluation loss achieved by Data-Parallel and DiLoCo for each model size $N$, along with percentage differences relative to Data-Parallel, in Figure 1. We see that for all values of $M$, the percentage difference strictly decreases with $N$. At $N = 2.4B$, DiLoCo with $M = 1$ or $2$ performs better than Data-Parallel. We validate this by training 4B and 10B models with hyperparameters set via our scaling laws. The results are in Table 4. We see that at 4B and 10B scales, DiLoCo with $M = 1, 2$ continue to do better than Data-Parallel. We discuss the scaling laws in detail in Section 6.

Table 4: Evaluation results on 4B and 10B models, using hyperparameters predicted by scaling laws. We indicate settings where DiLoCo reaches lower loss than Data-Parallel in bold.

| Algorithm | Loss | |
|---|---|---|
| | 4B | 10B |
| Data-Parallel | 2.224 | 2.090 |
| DiLoCo, $M = 1$ | **2.219 (-0.22%)** | **2.086 (-0.19%)** |
| DiLoCo, $M = 2$ | **2.220 (-0.18%)** | **2.086 (-0.19%)** |
| DiLoCo, $M = 4$ | 2.230 (+0.18%) | 2.096 (+0.29%) |

**Finding 2: DiLoCo with $M = 1$ attains lower evaluation loss than Data-Parallel across model scales.** DiLoCo with $M = 1$ also achieves higher downstream zero-shot accuracy, as we show in Figure 2. These plots also show that DiLoCo with $M = 1$ also exhibits greater stability with respect to batch size; doubling or quadrupling the batch size greatly reduced performance of Data-Parallel, but had little effect on DiLoCo, $M = 1$, as depicted in Figure 2.

**Finding 3: DiLoCo increases optimal batch size.** We plot evaluation loss as a function of batch size in Figure 3. Optimal batch size increases when using DiLoCo, and subsequently increases with $M$. As batch size increases, Data-Parallel becomes worse than DiLoCo with $M = 2, 4$, and eventually, $M = 8$. We show that the same occurs for downstream tasks in the appendix (see Figure 10). This means that DiLoCo exhibits more *horizontal scalability*, reducing training time via resource parallelization in addition to reducing communication. To show this, we plot an idealized wall-clock time when training under networks of varying bandwidth in Figure 4. DiLoCo's tolerance for larger batch sizes allows it to achieve comparable loss to Data-Parallel significantly faster.

**Finding 4: Optimal outer learning rate is constant with $N$.** While optimal inner learning rate varies with model size $N$, the optimal outer learning rate $\eta$ for DiLoCo is independent of $N$ and depends only on $M$. As shown in Figure 5, for sufficiently large models ($N \geq 335M$), the best $\eta$ for each $M$ is constant. Larger values of $M$ seem to necessitate larger $\eta$. This is consistent with

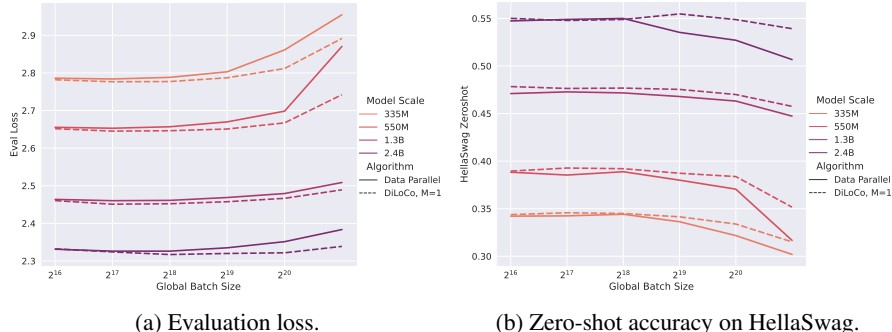

(a) Evaluation loss.
(b) Zero-shot accuracy on HellaSwag.

Figure 2: Evaluation loss and downstream accuracy of Data-Parallel and DiLoCo with $M = 1$ for varying model and global batch sizes (measured in tokens).

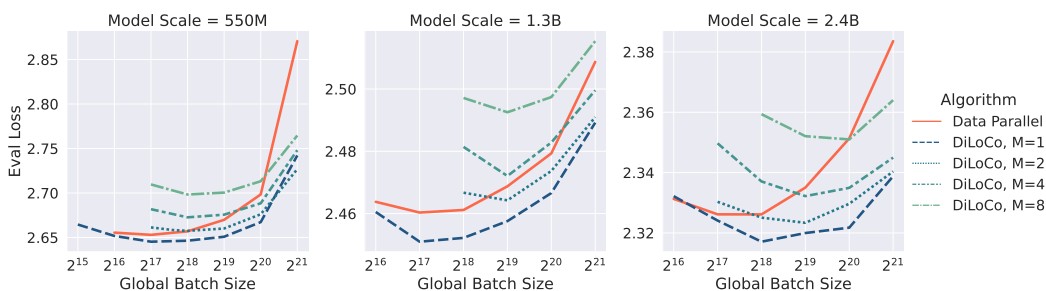

Figure 3: Evaluation loss of Data-Parallel and DiLoCo as a function of global batch size (in tokens). We see similar results for other model sizes, but omit for conciseness.

prior findings that outer learning rate should increase as a function of number of clients in federated learning settings [6].

## 5 Ablations

**Synchronization cadence.** Our experiments above all use a synchronization cadence $H$ of 30. We perform an ablation over $H$ for varying $M$ and model sizes $N$. For each setting, we take the optimal inner learning rate and global batch size from above, and sweep $H$ over $\{1, 5, 10, 30, 100, 300\}$ and

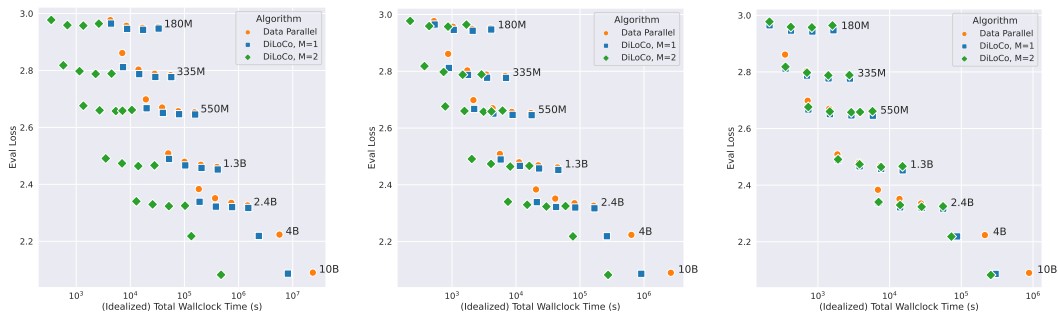

(a) Network with a bandwidth of 10 gigabits/s and a latency of $10^{-2}$ seconds (**low-bandwidth**).

(b) Network with a bandwidth of 100 gigabits/s and a latency of $10^{-3}$ seconds (**medium-bandwidth**).

(c) Network with a bandwidth of 400 gigabits/s and a latency of $10^{-4}$ seconds (**high-bandwidth**).

Figure 4: Idealized wall-clock time (see Section B) when training with Data-Parallel and DiLoCo across compute nodes connected via high-, medium-, and low-bandwidth networks, for varying model sizes. For models up to 2.4B, we also vary global batch size. For 4B and 10B models, we use the batch size predicted by scaling laws (see Section 6).

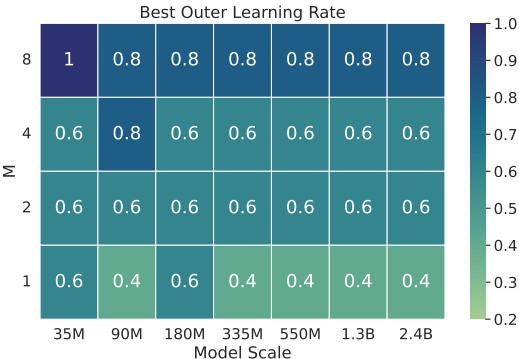

Figure 5: The best outer learning rate for DiLoCo for varying $M$ and model size $N$. We select the best outer learning rate over $\{0.2, 0.4, 0.6, 0.8, 1.0\}$, optimizing over inner learning rate $\gamma$ and global batch size $B$. For sufficiently large models, the best outer learning rate is constant.

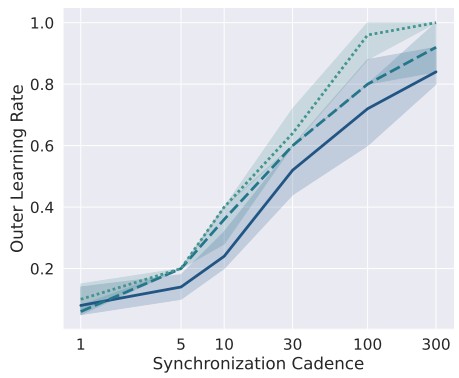

(a) Optimal outer learning rate for each synchronization cadence. Shaded regions represent the variance across model sizes.

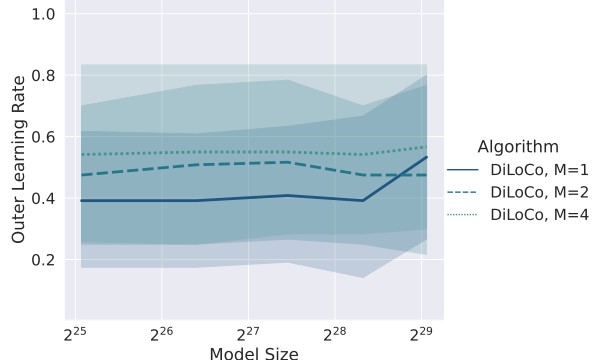

(b) Optimal outer learning rate for each model size. Shaded regions represent the variance across synchronization cadences.

Figure 6: Optimal outer learning rate $\eta$ as a function of synchronization cadence $H$ and model size $N$. The optimal $\eta$ increases with $H$ and $M$ (left), but is independent of model size $N$ (right).

$\eta$ over $\{0.05, 0.1, 0.2, 0.4, 0.6, 0.8, 1.0\}$. We first study whether the observation in Section 4, that $\eta$ should be tuned independently of $N$, holds for other values of $H$. We give results in the affirmative in Figure 6. We see that across model scales, the optimal learning rate is essentially only a function of the number of replicas $M$ and the synchronization cadence $H$, and is essentially independent of model size $N$. There is some slight variation, though this is likely due to not re-tuning the inner learning rate. Moreover, our results actually show a potentially counter-intuitive phenomenon: The optimal outer learning rate *increases* with $H$, even though the outer gradients increase in size as $H$ increases. We present additional analyses of synchronization cadence, including how $H$ impacts evaluation loss and compute utilization, in Section E.1.

**Overtraining.** Above, we used the Chinchilla-optimal amount of tokens for each model size [20]. It is often beneficial to perform *overtraining*, using more tokens than that [17]. We conduct ablations on various *overtraining multipliers*. Given an overtraining multiplier $\zeta \geq 1$, we train on $D = 20N\zeta$ tokens, so that $\zeta = 1$ corresponds to the Chinchilla-optimal number of tokens. Our results are in Figure 7. Qualitatively, the scaling remains essentially unchanged as we do more overtraining. We did not re-tune any hyperparameter in these experiments. For each model size and algorithm, we simply took the best-performing hyperparameters from our Chinchilla-optimal experiments on C4. This means that the consistency of DiLoCo as we overtrain held despite the fact that for $M > 1$, we used larger batch sizes than Data-Parallel. We shows this in detail in Section E.2.

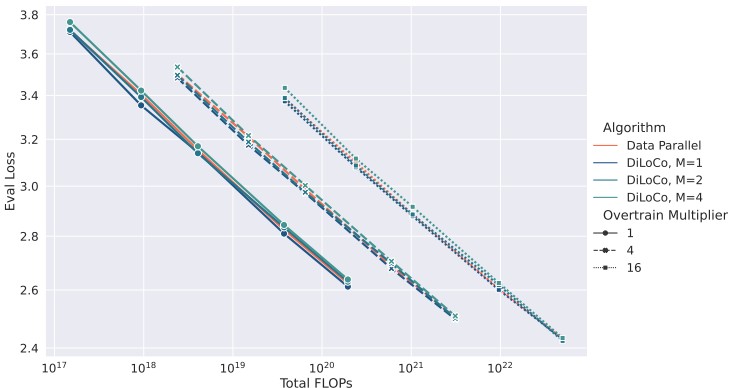

Figure 7: Evaluation loss as a function of FLOPs, for varying algorithms and amounts of over-training. For each overtrain multiplier, the curves are all essentially parallel lines.

## 6  Scaling Laws

We now discuss the process used to fit scaling laws to our empirical results. For each algorithm (Data-Parallel or DiLoCo with $M \in \{1, 2, 4, 8\}$) we ran extensive hyperparameter sweeps on models of size $N$ up to 2.4B. To fit a scaling law for loss $L$ as a function of $N$, we pick the best hyperparameters for each $N$, and fit a power law to $L(N) \approx AN^{\alpha}$. For DiLoCo, we fit two types of scaling laws: independent power law for each value of $M$ and joint fits where we fit a single scaling law $L(N, M) \approx AN^{\alpha}M^{\beta}$. We also fit power laws, both individual and joint, to the optimal learning rate $\gamma$ and batch size $B$. Due to a lack of space, we give the parameters of the power laws fit from our data in Section F. Here we overview some of the important findings derived from these scaling laws.

**Interpolation.**   First, we measured whether individual or joint fit scaling laws *interpolated* to our data in DiLoCo better. We do this via leave-one-out validation. We fit scaling laws for $L$, $\gamma$, and $B$, but only using data up to $N = 1.3B$ parameters, leaving out our data on $N = 2.4B$ parameters. We then use the scaling law to predict the optimal value for $L, \gamma$, and $B$ at $N = 2.4B$ parameters, across different values of $M$. We measure the residual $\text{res}(y, \tilde{y}) = |\log(y) - \log(\tilde{y})|$ between our prediction $\tilde{y}$ and the actual $y$, and average them across $M$. The results are in Table 5. We see that the joint fit matches independent for $L$ and $B$, but does better at predicting $\gamma$.

Table 5: Residuals for scaling law predictions at $N = 2.4B$ and varying $M$. We compare independent and joint scaling laws in predicting loss $L$, inner learning rate $\gamma$, and global batch size $B$. For the average residuals, we highlight which of independent or joint achieved a lower residual.

| | Fit | $L$ | $\gamma$ | $B$ |
|---|---|---|---|---|
| $M = 1$ | Independent | 0.011 | 0.35 | 0.00088 |
| | Joint | 0.019 | 0.14 | 0.19 |
| $M = 2$ | Independent | 0.0099 | 0.18 | 0.44 |
| | Joint | 0.013 | 0.29 | 0.28 |
| $M = 4$ | Independent | 0.012 | 0.051 | 0.25 |
| | Joint | 0.0082 | 0.086 | 0.11 |
| $M = 8$ | Independent | 0.014 | 0.62 | 0.076 |
| | Joint | 0.0076 | 0.23 | 0.19 |
| Average over $M$ | Independent | **0.012** | 0.30 | **0.19** |
| | Joint | **0.012** | **0.19** | **0.19** |

**Extrapolation.**   Next, we use the scaling laws for hyperparameters to predict optimal hyperparameters for $N = 4B$ and 10B, for Data-Parallel and DiLoCo with $M \in \{1, 2, 4\}$. We train using those

hyperparameters, and measure how well our scaling laws extrapolated. We compare the evaluation loss of these models in Table 6. We also show how well these aligned with our actual scaling laws for loss in Figure 8. Our extrapolation reveals a few key points. First, the trends discussed in Section 4 continue to hold at larger model scales, where DiLoCo with $M = 1, 2$ does better than Data-Parallel. Second, as with our interpolation experiments, joint-fit power laws extrapolate better for DiLoCo.

Table 6: Evaluation results on 4B and 10B models, using hyperparameters predicted by individual and joint scaling laws. We highlight DiLoCo evaluation results that were better (ie. lower) than Data-Parallel. We see that while DiLoCo with $M = 2$ does better than Data-Parallel with either independent or joint fit rules, DiLoCo $M = 1$ only does better when using joint fit.

| Algorithm | Fit Method | Loss | |
| --- | --- | --- | --- |
| | | 4B | 10B |
| Data-Parallel | Independent | 2.224 | 2.090 |
| DiLoCo, $M = 1$ | Independent | 2.229 | 2.103 |
| | Joint | **2.219** | **2.086** |
| DiLoCo, $M = 2$ | Independent | **2.218** | **2.083** |
| | Joint | **2.220** | **2.086** |
| DiLoCo, $M = 4$ | Independent | 2.232 | 2.098 |
| | Joint | 2.230 | 2.096 |

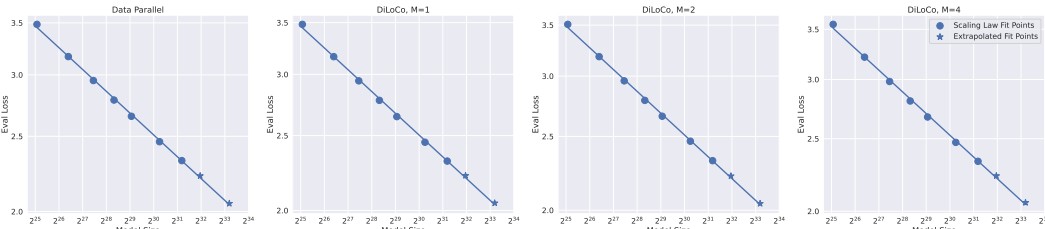

Figure 8: Scaling laws for Data-Parallel and DiLoCo. Pictured are both the loss values to form the scaling law (by training models up to a scale of 2.4B) and loss values attained on larger models (4B and 10B). While we present the individual-fit scaling laws for simplicity (and the ability to visualize Data-Parallel), the joint fit also predicts loss similarly well.

**Other results.**  Due to a lack of space, we relegate other findings of our scaling laws to Section F. There we discuss the methodology in more detail, as well as the actual parameters of the power laws. We also discuss fitting other functional forms to the scaling laws, beyond simple power laws.

## 7  Discussion and Limitations

Our results above all show that like Data-Parallel, DiLoCo scales predictably with model size in ways that make it simpler to tune hyperparameters and train models at extremely large scales. Moreover, DiLoCo can offer significant benefits over Data-Parallel, including superior evaluation loss when using a single model replica, and increased optimal batch size for any number of model replicas. These benefits are robust to model scale, overtraining amount, and synchronization frequency.

While promising, there are clear limitations and directions for future exploration. First, while we have done our best to include downstream evaluations on our models (Section D), careful evaluation across domains and downstream tasks is important for validating scaling behavior. Moreover, there are a wide variety of datasets used for pre-training. While we have incorporated two widely used datasets of this kind in our analysis, we have not validated robustness across datasets. Second, like Hoffmann et al. [20], we use dense transformer architectures of varying sizes, and have not validated these results on alternative architectures (such as Mixture-of-Experts) or in other domains, notably multi-modal settings. Third, we use an idealized wall-clock model (Section B), leaving to future work a thorough analysis of runtime in practical systems, and how it varies according to the type of compute. Last,

while we have attempted to be comprehensive in analyzing the core DiLoCo algorithm, our work does not encompass various related methods, and does not constitute a comprehensive algorithmic benchmark. While we expect our findings transfer to related methods, explicit verification of this is left to future work.

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

# A   Related Work

**Distributed training of LLMs.**   Due to their increasingly large sizes, the advancement of language models has necessitated advancements in distributed training methods. One vein of work uses data-parallel training, and attempts to shard its constituent computations across accelerators in efficient ways. This includes advancements in things like distributed data parallelism [52, 28], ZeRO parallelism [46, 48], fully-sharded data parallelism [15, 63], and pipeline parallelism [44, 22, 39]. Conceptual models of the impact of batch size on training time [35] aid in making the most effective trade-offs between training time and compute cost when using data-parallel training methods.

As scale continues to increase, the need to all-reduce gradients between data-parallel replicas becomes a bottleneck in training, causing accelerators to 'wait' on this allreduce for an unacceptably long time. This observation has motivated extensive work in pipeline parallelism and scheduling, communicating activations rather than gradients. An alternative line of work keeps the basic data processing pattern of data parallelism while directly minimizing communication requirements.

Three broad families of algorithms exist in that space: (1) sparse updates (including CocktailSGD [58], PowerSGD [56], DeMo [43], and Dion [1]), (2) fast asynchronous updates (including Hogwild [40], WASH [16], and Sparta [3]), and (3) infrequent updates (including LocalSGD [54], FedOpt [47], DiLoCo [11], and PALSGD [38]). These are extremely active areas of work, and we do not attempt to give a comprehensive survey of them all simultaneously. We instead defer the interested reader to a survey of decentralized LLM training [10].

In this work, we focus on the third category. Douillard et al. [11] showed that we can reduce communication costs in LLM training significantly by training multiple models independently with infrequent synchronization. Their method, DiLoCo, massively reduces communication overhead when training LLMs with a moderate numbers of model replicas. It has also shown great promise in training LLMs up to 10 billion parameters [24, 23]. This work has also been extended to asynchronous overlapped updates [30, 13], and low-communication expert sharding [12].

**Federated learning.**   There is an enormous body of work on communication-efficient training methods for machine learning. In that vein, DiLoCo is closely related to algorithms used in federated learning to perform communication-efficient training over decentralized data, often (but not exclusively) on edge devices [25]. The prototypical algorithm used in federated learning, FedAvg [36], reduces communication costs by training models in parallel, with periodic model averaging. This algorithm has been invented and reinvented throughout machine learning, and is also known as Local SGD [54], parallel SGD [64], and parallel online backpropagation [34]. The use of inner and outer optimization steps (as in DiLoCo, see Algorithm 1) was first used by Hsu et al. [21] and Reddi et al. [47] for federated learning, focusing on SGD as the inner optimizer and SGDM or Adam [27] as the outer optimizer in order to leverage more sophisticated optimizers in resource-constrained settings. DiLoCo is also closely related to many other federated optimization methods, though the huge amount of work in this area makes it impossible to summarize succinctly here. We instead refer the interested reader to the survey of Wang et al. [57], though the field has of course progressed since then. While federated learning is often applied to more moderately sized models, Charles et al. [7] and Sani et al. [50] show that federated learning can be used to good effect for LLM training.

**Scaling laws.**   Scaling laws work often aims to estimate how empirical generalization error scales with various facets, including model size and training set size. Empirical scaling analyses with power law behavior date have existed for decades (see [2]). Hestness et al. [19] developed power laws for model and dataset size across various tasks and model architectures (including encoder-decoder LSTM models). More recently, scaling laws for transformer-based LLMs were proposed by Kaplan et al. [26] and Hoffmann et al. [20], who exhibited power law relationships between LLM performance and model size. Sine then, there has been a large number of works developing scaling laws for other facets of LLMs, including (among many others) inference costs [51], data-constrained training [37], and overtraining [17].

Scaling laws for DiLoCo were previously studied by He et al. [18], who show that DiLoCo with 8 replicas exhibits analogous scaling behavior to Data-Parallel. He et al. [18] use a fixed number of replicas, batch size, and outer learning rate, and an unspecified total token budget.[4] Our work

---

[4]At the time of writing, He et al. [18] only say that "Each model was trained to achieve adequate convergence".

expands on many aspects of their work and explores others that were not considered, including but not limited to: $10\times$ larger models, varying the number of replicas (including single-replica DiLoCo), varying token budgets and overtraining, parametric function fitting, scaling laws for hyperparameters, and optimal batch size analysis.

# B   Wall-Clock Time Model

In this section, we present an idealized model for the wall-clock times of Data-Parallel and DiLoCo. We measure the total elapsed time, which means that parallelization (e.g. via increasing the batch size) reduces wall-clock time.

## B.1   Computation Time

Here, we mean the time expended by floating point operations in model training, ignoring communication time across nodes (which we treat separately in the section below). We use the idealized model where the total FLOPs $C = 6ND$. Given some number of chips $R$, each of which can perform $Q$ floating point operations per second, the total computation time is bounded below by $C/RQ$. The number of chips $R$ is purely a function of model size $N$ and global batch size $B$. The number of chips does not depend on the algorithm (Data-Parallel or DiLoCo) or number of model replicas when using DiLoCo.

## B.2   Communication Time

The network connectivity is characterized by a bandwidth $W$ and latency $\epsilon$. When performing an all-reduce of $N$ parameters over $R$ compute nodes, the lower bound on the amount of traffic sent and received by at least one of the compute nodes participating in the all-reduce is $2N(1 - R^{-1})$ [42]. Such algorithms are called *bandwidth-optimal*. Since communication across the nodes is done synchronously but in parallel, in a network with bandwidth $W$ and latency $\varepsilon$ between each pair of nodes, the time to complete the all-reduce is at least

$$\frac{2N}{W}\left(1 - \frac{1}{R}\right) + \varepsilon.$$

DiLoCo [11] was designed for settings where models are too large to fit in a single datacenter, so they must be trained across compute islands connected by low bandwidth. To model this, we will assume that we are training over $R$ compute nodes (typically, GPUs or TPUs). Some of these are connected by networks within a datacenter, and others are connected across datacenters. We let $W_0, \varepsilon_0$ denote the bandwidth and latency of the within-datacenter network, and $W_1, \varepsilon_1$ analogously defined for the cross-datacenter network. Typically, $W_0 \geq W_1, \varepsilon_0 \leq \varepsilon_1$.

**Data-Parallel:**   At every training step $T$, we have to perform an all-reduce over all $R$ compute nodes. Since some nodes are connected across datacenters, the total communication time is at least

$$\left(\frac{2N}{W_1}\left(1 - \frac{1}{R}\right) + \varepsilon_1\right)T$$

**DiLoCo, $M = 1$:**   At every inner step $T$, we perform an all-reduce over all $R$ devices as in Data-Parallel training. We also do an all-reduce every $H$ steps for the outer optimization. Some of these nodes are connected across datacenters, so the communication time per all-reduce is at least $2N(1 - R^{-1})W_1^{-1} + \varepsilon_1$. The total communication time is therefore at least

$$\left(\frac{2N}{W_1}\left(1 - \frac{1}{R}\right) + \varepsilon_1\right)T\left(1 + \frac{1}{H}\right).$$

**DiLoCo, $M \geq 2$:**   We assume that each of the $M$ model replicas is trained on compute nodes connected within the same datacenter. In each inner step $T$, each model replica is trained by $R/M$ devices which need to do an all-reduce. However, no communication occurs between datacenters, so the communication time of each inner step is bounded by $2N(1 - MR^{-1})W_0^{-1} + \varepsilon_0$.

Each outer optimization step involves all-reducing over all $R$ devices, connected across datacenters. This incurs a communication time of at least $2N(1 - R^{-1})W_1^{-1} + \varepsilon_1$. Since it occurs only every $H$ steps, the total communication time is bounded below by:

$$\left(\frac{2N}{W_0}\left(1 - \frac{M}{R}\right) + \varepsilon_0\right)T + \left(\frac{2N}{W_1}\left(1 - \frac{1}{R}\right) + \varepsilon_1\right)\frac{T}{H}.$$

Note that this suggests that as long as $H \geq W_0/W_1$, the outer communication steps incur at most half of the total communication cost.

**Streaming DiLoCo.** We note that the computed cost above applies to the Streaming DiLoCo [13] as well. While the inner step remains the same, the outer step is smoothed such that each fragment $p \in \{1, \ldots, P\}$ is every $H$ steps. However, fragment communication is offset such that some fragment is communicated every $H/P$ steps, resulting in the communication amortizing to the same per-step cost. This is expected as Streaming DiLoCo reduces peak communication over any step, but does not reduce total communication across training.

**Overlapping communications.** Another contribution of Douillard et al. [13] is the ability to overlap communications required for the outer optimizer with computation by using a stale version of the fragment in the outer optimizer, and merging the result of this outer optimization with the locally optimized fragment. This would allow, for example, the communication term to be omitted from the calculation for wall-clock-time, if computation time dominates communication time. This setting is different from an algorithmic perspective, so its impact on scaling would need to be examined independently.

### B.3 Total Wall-Clock Time

The total wall-clock time is a sum of the computation and communication times above. To measure the communication time, we must know the number of chips $R$ used for each experiment, the number of FLOPs per chip per second $Q$, the bandwidth and the latency of the within-datacenter and cross-datacenter networks. For computation costs, we use a slightly idealized number of chips $R$ based on our experiments, but ensuring that doubling the global batch size would double the number of chips. We base the constant $Q$ on publicly available information about the FLOPs capabilities of the TPU v5e and v6e chips[5], which have peak compute per chip (in bfloat16) of 197 teraflops and 918 teraflops, respectively. Assuming a maximum FLOPs usage of 50%, these chips have an actual compute of approximately 100 and 408 teraflops, respectively. When computing idealized compute time, we set $Q = 300$ teraflops, somewhere in-between the two.

For bandwidth and latency, we consider three archetypes of networks:

1. **High-bandwidth network** with bandwidth $W_{\text{high}} = 400$ gigabits per second and a latency of $\varepsilon_{\text{high}} = 10^{-4}$ seconds.

2. **Medium-bandwidth network** with bandwidth $W_{\text{med}} = 100$ gigabits per second and a latency of $\varepsilon_{\text{med}} = 10^{-3}$ seconds.

3. **Low-bandwidth network** with bandwidth $W_{\text{low}} = 10$ gigabits per second and a latency of $\varepsilon_{\text{low}} = 10^{-2}$ seconds.

We stress that these are not based on any actual systems, and are simply designed as instructive archetypes of networks. For the idealized communication time, we always use the high-bandwidth network for the within-datacenter network, and one of the three for the cross-datacenter network.

## C  Datasets and Licenses

In this section, we discuss the datasets used in our experiments and their respective licenses. We note that the datasets themselves are all referenced in Section 3.

---

[5]See https://cloud.google.com/tpu/docs/v6e.

- C4 [45]. This dataset is a cleaned version of Common Crawl's web crawl corpus[6]. We used the version accessible via Hugging Face, provided by Allen AI[7], and made available under the ODC-BY license. Because it is derived from Common Crawl, its usage is also bound by the Common Crawl terms of use. We use the **en** version of the dataset. We note that while widely used and filtered, the dataset still contains undesirable content including hate speech. See [33] for an analysis of its contents, and discussion of mitigation strategies. We do not release any assets derived from this dataset.

- Dolma [53]. This dataset consists of 3 trillion tokens drawn from web content, academic publications, code, books, and related material. We used the **v1_7** version of the dataset accessible via Hugging Face, provided by Allen AI[8], and made available under the ODC-BY license. We note that the usage of this dataset is also bound by license agreements and restrictions of its original data sources. See [53] for more details.

- HellaSwag [61]. We use this dataset for zero-shot evaluation metrics. We used the version available under the MIT license[9] For details, see `https://rowanzellers.com/hellaswag/`.

- Piqa [4]. We use this dataset for zero-shot evaluation metrics. We used the version available under the Academic Free License v3.0[10].

- Arc-Easy [9]. We use the Arc-Easy dataset for zero-shot evaluation metrics. We use the version accessible via Hugging Face, provided by Allen AI[11], made available under the CC-BY-SA-4.0 license.

## D  Additional Experimental Results

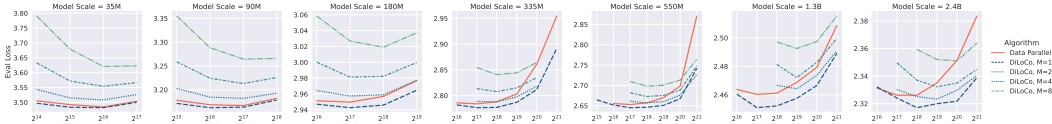

Figure 9: Evaluation loss of Data-Parallel and DiLoCo as a function of global batch size (in tokens).

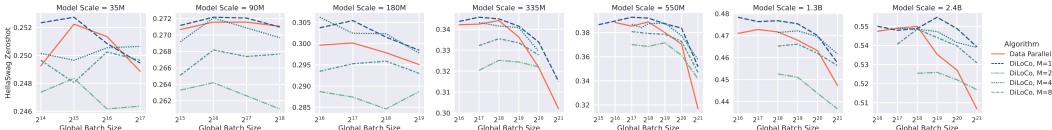

Figure 10: Zero-shot evaluation accuracy on HellaSwag of Data-Parallel and DiLoCo as a function of global batch size (in tokens).

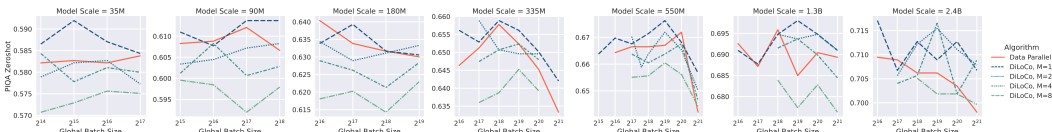

Figure 11: Zero-shot evaluation accuracy on Piqa of Data-Parallel and DiLoCo as a function of global batch size (in tokens).

In this section, we give additional experimental results that expand on those in Section 4. In Figures 9, 10, 11, and 12, we present evaluation loss and evaluation accuracy on various downstream zero-shot

---

[6]`https://https://commoncrawl.org/`

[7]`https://huggingface.co/datasets/allenai/c4`

[8]`https://huggingface.co/datasets/allenai/dolma`

[9]`https://github.com/rowanz/hellaswag`.

[10]`https://github.com/ybisk/ybisk.github.io/tree/master/piqa`

[11]`https://huggingface.co/datasets/allenai/ai2_arc`

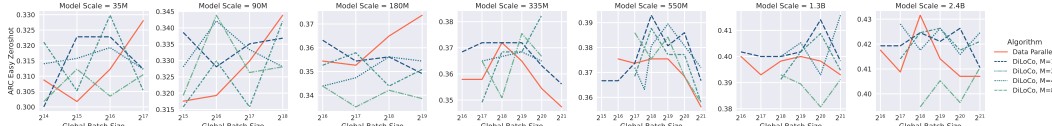

Figure 12: Zero-shot evaluation accuracy on Arc-Easy of Data-Parallel and DiLoCo as a function of global batch size (in tokens).

tasks, as a function of algorithm, model size, and global batch size. The results consistently show that Data-Parallel's evaluation performance degrades quickly as batch size increases. By contrast DiLoCo's performance degrades more slowly, or even improves, as batch size increases. We note that Arc-Easy was quite noisy as an evaluation task.

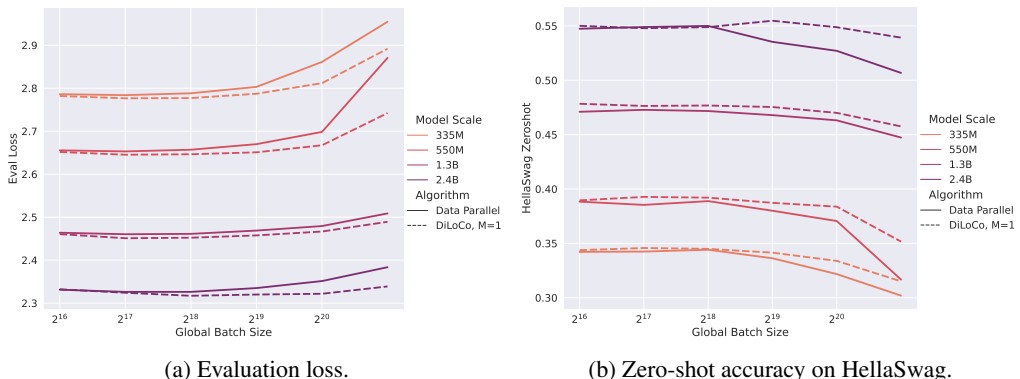

(a) Evaluation loss.

(b) Zero-shot accuracy on HellaSwag.

Figure 13: Evaluation loss and zero-shot accuracy of Data-Parallel and DiLoCo with $M = 1$ for varying model and global batch sizes (measured in tokens). In all settings, DiLoCo with $M = 1$ does better than Data-Parallel, and the gap between them increases with batch size.

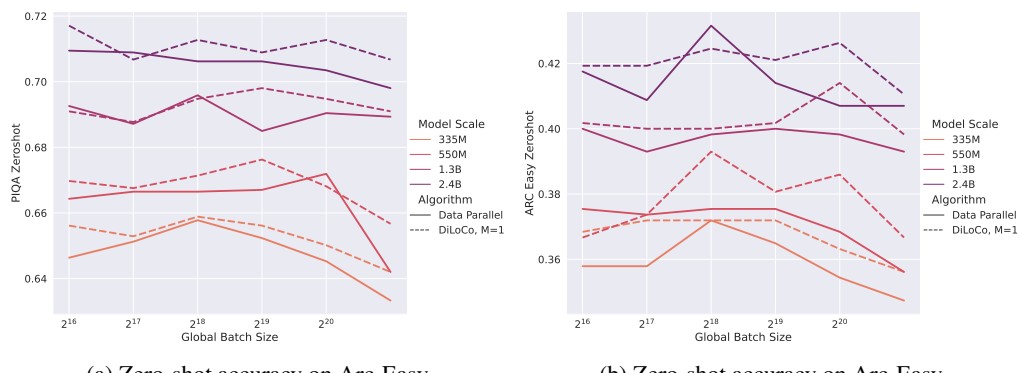

(a) Zero-shot accuracy on Arc-Easy.

(b) Zero-shot accuracy on Arc-Easy.

Figure 14: Zero-shot evaluation accuracy of Data-Parallel and DiLoCo with $M = 1$ for varying model and global batch sizes (measured in tokens), on Piqa and Arc-Easy. In nearly all settings, DiLoCo with $M = 1$ does better than Data-Parallel, and the often the gap increases with batch size.

In Figures 13 and 14, we compare Data-Parallel and DiLoCo with $M = 1$ in terms of their evaluation loss and zero-shot evaluation accuracy on HellaSwag, Piqa and Arc-Easy. As above, we note that DiLoCo with $M = 1$ has an improved tolerance to larger batch sizes.

# E    Ablations

Here, we expand upon the ablation studies discussed in Section 5. We give more detailed experimental analyses, as well as idealized wall-clock times under varying networks.

## E.1 Synchronization Cadence

As discussed in the synchronization cadence ablations in Section 5, we perform an ablation over $H$ for varying $M$ and model sizes $N$. For each setting, we take the optimal inner learning rate and global batch size from above, and sweep $H$ over $\{1, 5, 10, 30, 100, 300\}$ and $\eta$ over $\{0.05, 0.1, 0.2, 0.4, 0.6, 0.8, 1.0\}$.

In addition to the results in Section 5, we analyze how $H$ impacts the evaluation loss of DiLoCo in Figure 15. For all models, synchronizing every step ($H = 1$) performs the worst, but after this point all values of $H$ perform somewhat comparably. For a fixed $N$ and $M$, evaluation loss increases as $H$ increases. However, this increase is less pronounced for $M = 1$ and larger models. This yields an important finding: As the model size $N$ increases, we can actually perform synchronization across DiLoCo replicas less frequently, while nearly maintaining evaluation performance.

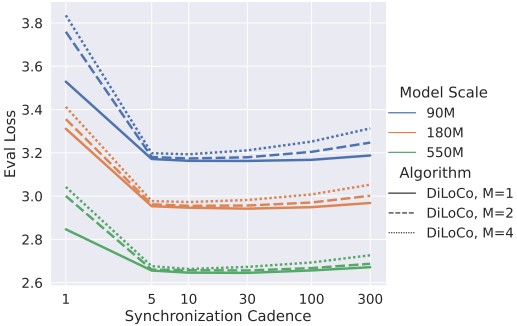

Figure 15: **Infrequent synchronization works better for larger models.** Outside of $H = 1$, which performs the worst, evaluation loss increases with $H$. However, the rate of increase is less pronounced for DiLoCo with $M = 1$ and for larger models, suggesting that large models can be synchronized quite infrequently.

Table 7: **Simulated compute utilization**. We estimate the step time based on the required flops using the rule proposed by Kaplan et al. [26] and a max flop utilization of 60%. We estimate the bandwidth (in Gbit/s) required to reach a level of compute utilization using [13]'s simulator. We highlight in light blue 10× reduction of bandwidth, and in dark blue 100× reduction.

| Architecture | Size | Step time | Method | Gbit/s to reach a compute utilization CU =? | | | | |
| | | | | 50% | 80% | 90% | 95% | 99% |
|---|---|---|---|---|---|---|---|---|
| Chinchilla | 10B | 0.8s | Data-Parallel | 104.8 | 184.2 | 222.3 | 222.3 | 390.7 |
| | | | DiLoCo, $H = 1$ | 104.8 | 184.2 | 222.3 | 222.3 | 390.7 |
| | | | DiLoCo, $H = 10$ | 16.0 | 49.4 | 86.8 | 152.6 | 222.3 |
| | | | DiLoCo, $H = 50$ | **3.0** | **11.0** | 23.3 | 41.0 | 126.5 |
| | | | DiLoCo, $H = 100$ | **1.4** | **6.2** | **13.3** | 23.3 | 86.8 |
| | | | DiLoCo, $H = 300$ | **0.5** | **2.0** | **4.3** | **9.1** | 41.0 |
| Llama3 | 405B | 26s | Data-Parallel | 126.5 | 222.3 | 268.3 | 323.8 | 323.8 |
| | | | DiLoCo, $H = 1$ | 126.5 | 222.3 | 268.3 | 323.8 | 323.8 |
| | | | DiLoCo, $H = 10$ | 19.3 | 72.0 | 126.5 | 184.2 | 268.3 |
| | | | DiLoCo, $H = 50$ | **3.6** | **13.3** | 28.1 | 59.6 | 184.2 |
| | | | DiLoCo, $H = 100$ | **2.0** | **7.5** | **16.0** | 33.9 | 126.5 |
| | | | DiLoCo, $H = 300$ | **0.7** | **3.0** | **6.2** | **13.3** | 59.6 |
| DeepSeek-V3 | 671B | 20s | Data-Parallel | 323.8 | 569.0 | 686.6 | 686.6 | 1000.0+ |
| | | | DiLoCo, $H = 1$ | 323.8 | 569.0 | 686.6 | 686.6 | 1000.0+ |
| | | | DiLoCo, $H = 10$ | 49.4 | 152.6 | 268.3 | 390.7 | 686.6 |
| | | | DiLoCo, $H = 50$ | **7.5** | **33.9** | 72.0 | 126.5 | 390.7 |
| | | | DiLoCo, $H = 100$ | **4.3** | **16.0** | **41.0** | 72.0 | 268.3 |
| | | | DiLoCo, $H = 300$ | **1.7** | **6.2** | **13.3** | **28.1** | 126.5 |

**Compute utilization.** The synchronization cadence of DiLoCo is critical to training large-scale models distributed across the world. Indeed, less frequent synchronization (larger $H$) diminishes

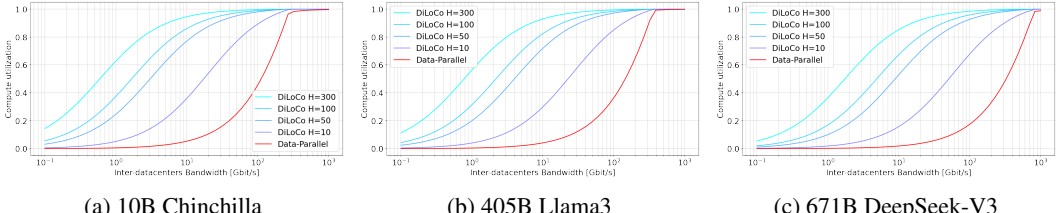

(a) 10B Chinchilla        (b) 405B Llama3        (c) 671B DeepSeek-V3

Figure 16: **DiLoCo greatly increases compute utilization**. Here we present simulated compute utilization for DiLoCo and Data-Parallel across a range of bandwidth and synchronization cadences $H$. A compute utilization of 0.8 means 80% of the time is spent in computation, and 20% in communication. A higher value is better. We see similar results for other model sizes, but omit for visual clarity.

the bandwidth requirements of training. Following [13], we simulate the amount of bandwidth required to have a compute utilization ($\frac{\text{compute time}}{\text{total time}}$) as large as possible for three types of LLMs: a 10B Chinchilla-style transformer [20] in Fig.16a, a 405B Llama3 model [14] in Fig.16b, and a 671B DeepSeek-v3 MoE [29] in Fig.16c. We also report raw numbers in Table 7.

## E.2 Overtraining

We expand on the overtraining ablation study in Section 5. As we showed there, DiLoCo continues to enjoy large optimal batch sizes, compared to Data-Parallel. This improves horizontal scalability, decreasing end-to-end wallclock time. Wallclock time is also smaller for $M > 1$ due to reduced communication.

To illustrate this, we plot idealized training time of Data-Parallel and DiLoCo with $M = 2$ for different overtraining amounts in Figure 17. DiLoCo speeds up overtraining by reducing communication costs, and utilizing larger batch sizes, therefore requiring fewer serial training steps. This suggests that DiLoCo is a significant boon in overtraining, as we can amortize compute time (which can be quite long for overtraining) via horizontal scalability.

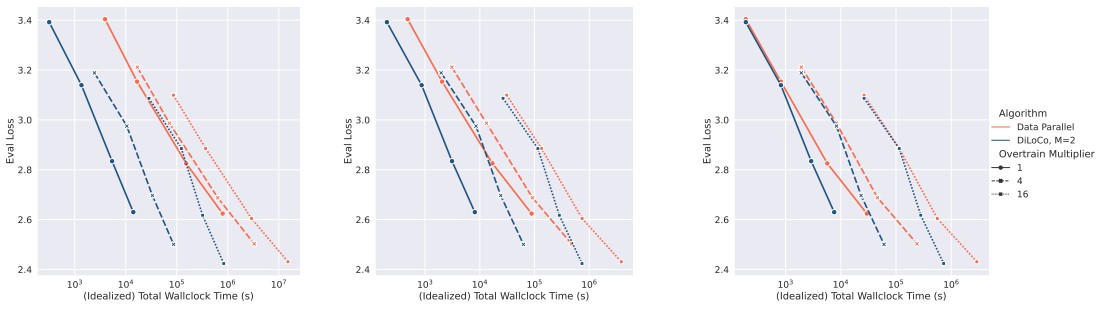

(a) Network with a bandwidth of 10 gigabits/s and a latency of $10^{-2}$ seconds (**low-bandwidth**).

(b) Network with a bandwidth of 100 gigabits/s and a latency of $10^{-3}$ seconds (**medium-bandiwdth**).

(c) Network with a bandwidth of 400 gigabits/s and a latency of $10^{-4}$ seconds (**high-bandwidth**).

Figure 17: For Data-Parallel and DiLoCo, $M = 2$, we plot idealized wall-clock time (see Section B) for training by Data-Parallel and DiLoCo, $M = 2$ across compute nodes connected via high-, medium-, and low-bandwidth networks. For each algorithm and overtraining amount, we display lines represent varying model sizes, from 335M parameters to 2.4B. DiLoCo is faster in all settings, due to both its reduced communication and its tolerance to larger batch sizes. Even in the high-bandwidth setting, the larger batch sizes increase horizontal scalability, reducing end-to-end wallclock time. We see similar results for $M \geq 4$ and for smaller models, but omit for visual clarity.

# F   Scaling Laws

We now discuss the process we used to fit scaling laws to our empirical results, expanding on the discussion in Section 6. Recall that for each algorithm (Data-Parallel or DiLoCo with $M \in \{1, 2, 4, 8\}$) we ran extensive hyperparameter sweeps on models of size $N$ up to 2.4B. To fit a scaling law for loss $L$ as a function of $N$, we pick the best hyperparameters (in terms of evaluation loss) for each $N$, aggregate the loss values, and fit some kind of function for $L(N)$, such as a power law [26]. We will also fit scaling laws to the optimal hyperparameters. We will fit two types of scaling laws for DiLoCo: independent fits for each value of $M$, and joint fits where we fit a single scaling law as a function of $N$ and $M$ simultaneously.

## F.1   Independent scaling laws

**Scaling laws for loss.**   We first fit scaling laws for the loss obtained by Data-Parallel training. We fit a power law to the evaluation loss of Data-Parallel as a function of $N$ via the power law approximation $L(N) \approx AN^\alpha$. Note that this can easily be done via applying linear fit techniques to $\log(L)$, and is not sensitive to things like initial values of $A, \alpha$. The resulting power law is in the first row of Table 8.

We mirror this above for DiLoCo when doing independent fits. For each value of $N, M$, we record the lowest loss value across all hyperparameters. We can then fit power law $L_M(N) := L(N, M) \approx AN^\alpha$ for each $M$. The results are given in Table 8.

Table 8: Power law approximations for loss $L(N) \approx AN^\alpha$.

|  | $A$ | $\alpha$ |
|---|---|---|
| Data-Parallel | 18.129 | $-0.0953$ |
| DiLoCo, $M = 1$ | 18.363 | $-0.0961$ |
| DiLoCo, $M = 2$ | 18.768 | $-0.0969$ |
| DiLoCo, $M = 4$ | 19.762 | $-0.0992$ |
| DiLoCo, $M = 8$ | 21.051 | $-0.1018$ |

The results show that Data-Parallel and DiLoCo see similar predicted reductions in loss as a function of $N$. Notably, the fit parameters suggest that DiLoCo, $M = 1$ outperforms Data-Parallel at essentially all but the absolute smallest model scales. This mirrors the results discussed in Section 4.

**Scaling laws for hyperparameters.**   For Data-Parallel, we fit scaling laws for learning rate $\gamma$ and batch size $B$. For DiLoCo, we fit scaling laws for inner learning rate $\gamma$ and global batch size $B$. Given their analogous role in the algorithms, we fit them in the same way. For (inner) learning rate, we use the same approach as fitting scaling laws for loss: for each $N$ (and $M$, for DiLoCo), we select the best hyperparameters, and fit a power law. The results are in Table 9.

For (global) batch size, we alter this slightly. As discussed in Section 3, our sweeps use powers of 2 for batch size, in order to saturate compute. However, the optimal batch size may be between these values. To account for this, we first fit a quadratic approximation to the batch size. Specifically, for each value of $N$ we look at the loss as a function of $\log_2(B)$ (when using the best learning rate for that $B$), and fit a quadratic to this function. We select the minima of those quadratics and fit a power law to them, as a function of $N$. The results are in Table 10.

DiLoCo has a third hyperparameter we could fit scaling laws to: the outer learning rate. However, as shown in Section 4, the optimal outer learning rate is (for sufficiently large models) seemingly constant. Therefore, a scaling law would seemingly not yield any improved predictive performance over simply using the best outer learning rate for each $M$ (see Figure 5).

## F.2   Joint scaling laws

Alternatively, we can fit joint power laws to various facets of DiLoCo, using a two-variable power law $f(N, M) \approx AN^\alpha M^\beta$. We do this for loss $L$, inner learning rate $\gamma$ and global batch size $B$. For the first two, we select, for each value of $N, M$, the best learning rate and loss. For batch size we do the same, but using the quadratic approximations from the section above. We can then fit a joint

Table 9: Power law approximations for (inner) learning rate $\gamma(N) \approx AN^\alpha$.

|  | $A$ | $\alpha$ |
|---|---|---|
| Data-Parallel | 16319.2 | $-0.819$ |
| DiLoCo, $M = 1$ | 74620.6 | $-0.945$ |
| DiLoCo, $M = 2$ | 3978.82 | $-0.780$ |
| DiLoCo, $M = 4$ | 4512.99 | $-0.789$ |
| DiLoCo, $M = 8$ | 618986 | $-1.102$ |

Table 10: Power law approximations for (global) batch size $B(N) \approx AN^\alpha$.

|  | $A$ | $\alpha$ |
|---|---|---|
| Data-Parallel | 462.68 | 0.281 |
| DiLoCo, $M = 1$ | 27.873 | 0.435 |
| DiLoCo, $M = 2$ | 15.749 | 0.479 |
| DiLoCo, $M = 4$, | 10.957 | 0.510 |
| DiLoCo, $M = 8$ | 38.072 | 0.455 |

Table 11: Joint power law approximations $f(N, M) = AN^\alpha M^\beta$ for the loss $L$, inner learning rate $\gamma$, and batch size $B$ of DiLoCo.

|  | $A$ | $\alpha$ | $\beta$ |
|---|---|---|---|
| $L$ | 19.226 | $-0.0985$ | 0.0116 |
| $\gamma$ | 22256 | $-0.8827$ | 0.2929 |
| $B$ | 14.521 | 0.4695 | 0.3399 |

power law via standard linear regression techniques. The resulting power laws are in Table 11. Just as with the independent fits, we do not attempt to fit scaling laws to the outer learning rate $\eta$, as the optimal value is independent of $N$.

### F.3 Measuring goodness-of-fit

Now that we have two different ways of developing scaling laws for DiLoCo, we can attempt to ask which one yields better predictions. First we do this via leave-one-out validation. Specifically, we use the same methodology as above to fit scaling laws for $L$, $\gamma$, and $B$, but only using data up to $N = 1.3$B parameters, leaving out our data on $N = 2.4$B parameters. We then use the scaling law to predict the optimal value for $L, \gamma$, and $B$ at $N = 2.4$B parameters, across different values of $M$.

Table 12: **Joint fit scaling laws match or beat independent fit.** Here we give the residuals for scaling law predictions for $N = 2.4$B and varying $M$. We compare the residual of independent and joint fitting strategies in predicting loss $L$, inner learning rate $\gamma$, and global batch size $B$. For the average residuals, we highlight which of independent or joint achieved a lower residual. We see that the joint fit matches independent for $L$ and $B$, but does better at predicting $\gamma$.

|  | Fit | $L$ | $\gamma$ | $B$ |
|---|---|---|---|---|
| $M = 1$ | Independent | 0.011 | 0.35 | 0.00088 |
|  | Joint | 0.019 | 0.14 | 0.19 |
| $M = 2$ | Independent | 0.0099 | 0.18 | 0.44 |
|  | Joint | 0.013 | 0.29 | 0.28 |
| $M = 4$ | Independent | 0.012 | 0.051 | 0.25 |
|  | Joint | 0.0082 | 0.086 | 0.11 |
| $M = 8$ | Independent | 0.014 | 0.62 | 0.076 |
|  | Joint | 0.0076 | 0.23 | 0.19 |
| Average over $M$ | Independent | **0.012** | 0.30 | **0.19** |
|  | Joint | **0.012** | **0.19** | **0.19** |

Given a predicted value $\tilde{y}$ and a reference value of $y$, we compute the *residual* of our prediction as the mean absolute error of the logarithm: $\mathrm{res}(y, \tilde{y}) = |\log(y) - \log(\tilde{y})|$. We use this measure as it works well for all three variables simultaneously, despite the fact that they vary greatly in their scale. For each $M \in \{1, 2, 4, 8\}$, we compute the predicted value of the three parameters above, and measure

the residual relative the actual optimal value at $N = 2.4B$. We report these values, as well as their average across $M$, in Table 12.

Our results generally show that both approach is generally valid (as there is no clear winner) but also that there is significant variation in residuals between $M$. That being said, we see that on average, while the individual fit is slightly better at predicting the loss and global batch size, the independent fit is significantly better at predicting inner learning rate.

### F.4 Extrapolating to larger models

We use the independent and joint fits to predict optimal hyperparameters for Data-Parallel and DiLoCo with $M \in \{1, 2, 4\}$ at 4B and 10B model scales. Note that for Data-Parallel, we can only use independent fits. We run training on these models with these hyperparameters, using a Chinchilla-optimal token budget, and compare the results.

Table 13: **Joint fit hyperparameters extrapolate well to larger models.** Here we show the evaluation results on 4B and 10B models, using hyperparameters predicted by individual and joint scaling laws. We highlight DiLoCo evaluation results that were better (ie. lower) than Data-Parallel. We see that while DiLoCo with $M = 2$ does better than Data-Parallel with either independent or joint fit rules, DiLoCo $M = 1$ only does better when using joint fit.

| Algorithm | Fit Method | Loss | |
|---|---|---|---|
| | | 4B | 10B |
| Data-Parallel | Independent | 2.224 | 2.090 |
| DiLoCo, $M = 1$ | Independent | 2.229 | 2.103 |
| | Joint | **2.219** | **2.086** |
| DiLoCo, $M = 2$ | Independent | **2.218** | **2.083** |
| | Joint | **2.220** | **2.086** |
| DiLoCo, $M = 4$ | Independent | 2.232 | 2.098 |
| | Joint | 2.230 | 2.096 |

We see two important facets. First, unlike results above, at 4B and 10B scales we see that DiLoCo with $M = 2$ actually outperforms both Data-Parallel and DiLoCo, $M = 1$, regardless of using individual or joint fit approaches. Second, we see that DiLoCo, $M = 1$ requires the joint fit to do better than Data-Parallel. Other than in this case, joint and independent fits perform comparably throughout. All in all, the joint fit approach to hyperparameters appears to have a slight edge over individual fit in extrapolating. Combined with its ability to also extrapolate to larger $M$, we generally recommend the joint fit approach for all hyperparameters.

We now use these loss values to see how they compare to the scaling laws fit above. We generally find that the loss values are predicted very well, within a few percentage points of the loss predicted by the scaling laws. We present the fit scaling law and extrapolated loss values in Figure 18.

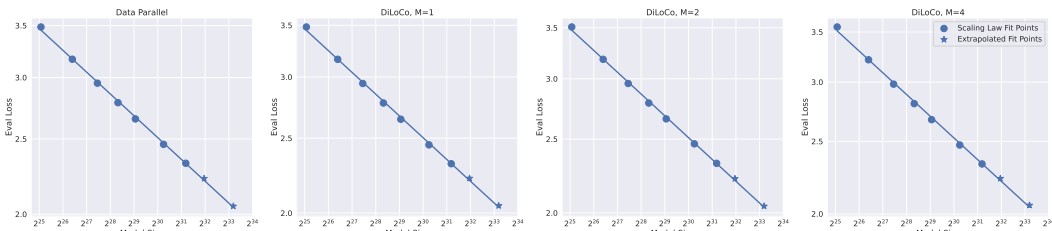

Figure 18: **DiLoCo scaling laws extrapolate well to larger models.** We present loss scaling laws for Data-Parallel and DiLoCo. Pictured are both the loss values to form the scaling law (by training models up to a scale of 2.4B) and loss values attained on larger models (4B and 10B). While we present the individual-fit scaling laws for simplicity, the joint fit also predicts loss well.

## F.5   Parametric Function Fitting

Various works on scaling laws have often found it useful to fit functions more complex than power laws to the data [20]. We are particularly interested in parametric forms for our joint scaling laws. While Hoffmann et al. [20] use a risk decomposition argument to decompose loss as a function of $N$ and $D$, it is not immediately clear how to do decompose $L(N, M)$. To that end, we use an empirical approach. We develop candidate functions, and determine which does best in an extrapolative sense. We use the following functional forms:

1. $L(N, M) \approx A N^\alpha M^\beta$
2. $L(N, M) \approx A N^\alpha M^\beta + C$
3. $L(N, M) \approx A N^{\alpha + \beta M} + C$
4. $L(N, M) \approx A N^\alpha + B M^\beta + C$

The first is included for comparison's sake, as it recovers the power law scaling law used above. Fitting more sophisticated functions can be much more sensitive to initial values of parameters, and also sensitive to outlier data. Therefore, when fitting these functions to our loss values above, we use the general strategy proposed by Hoffmann et al. [20].

In detail, let $f_Q(N, M)$ denote one of the functional forms above, where $Q$ represents the set of parameters to be fit (e.g. $Q = \{A, \alpha, \beta\}$ for the first). Let $\text{Huber}_\delta$ denote the Huber loss with parameter $\delta$. Let $\mathcal{N}, \mathcal{M}$ denote the set of values of $N$ and $M$ considered. For each $N, M$, we have an empirical loss $L(N, M)$, and some estimate of the loss $f_Q(N, M)$. We then solve the following minimization problem:

$$\min_Q \sum_{N \in \mathcal{N}} \sum_{M \in \mathcal{M}} \text{Huber}_\delta \bigg( \log f_Q(N, M) - \log L(N, M) \bigg).$$

We minimize this via L-BFGS, using some initialization $Q_0$ for the parameters. We repeat this process for 256 random initializations $Q_0$ of the parameters. We hold out all loss values at the $N$ = 2.4B scale, and select the parameters $Q$ that best fit the held-out data, measured in terms of the average residual $|\log f_Q(N, M) - \log L(N, m)|$ over all $M$.

Table 14: **Parametric function fitting improves joint scaling laws.** We showcase various parametric approximations to the empirical loss function $L(N, M)$, along with their validation error on held-out loss data at the $N$ = 2.4B scale. We see that joint power laws (the first) and classical loss decomposition (the last) are significantly worse at predicting loss on the held-out data.

| Parametric form | Average Residual |
|---|---|
| $L(N, M) \approx A N^\alpha M^\beta$ | 0.0044 |
| $L(N, M) \approx A N^\alpha M^\beta + C$ | 0.0035 |
| $L(N, M) \approx A N^{\alpha + \beta M} + C$ | **0.0025** |
| $L(N, M) \approx A N^\alpha + B M^\beta + C$ | 0.0043 |

We see that the power law (row 1) and additive decomposition (row 4) are significantly worse at extrapolating loss values than more nuanced parametric forms. We note that the additive decomposition resembles the decomposition of loss as a function of model size $N$ and token budget $D$ used by Hoffmann et al. [20], but does not seem to reflect how $M$ affects loss for DiLoCo. We leave it as an open problem to determine what parametric forms better predict loss, and can be explained by theoretical understanding of communication-efficient training.

