# OpenReview forum: "Communication-Efficient Language Model Training Scales Reliably and Robustly: Scaling Laws for DiLoCo"
_NeurIPS.cc/2025/Conference — NeurIPS 2025 spotlight_

### Official Review · Reviewer_6yd8 · 2025-06-17

**Clarity:** 3
**Significance:** 2
**Originality:** 2
**Rating:** 4
**Confidence:** 3

**Summary:**

This paper studied the scaling law of pretraining using DiLoCo and showed DiLoCo also follows the similar scaling law of standard DDP training along with a few emperical findings.

**Questions:**

At line 185 the authors mentioned an counter-intuitive observation: "outer learning rate increases with H". Do you have any potential explanation for it?

**Ethical Concerns:**

["NO or VERY MINOR ethics concerns only"]

**Final Justification:**

Convinced by the rebuttal that this work provide valuable insights to large scale LLM training.

**Limitations:**

as is mentioned earlier, the efficacy is limited to DiLoCo with low network bw, which is typically inter-datacenter and less practical for majority of ML practioners

**Quality:**

3

**Strengths And Weaknesses:**

Strength:

* The study is comprehensive, covering the emperical similarity to DDP as well as its own unique properties. It also explores the effect of 1st order hyperparameters like outer learning rate of DiLoCo, batch-size, etc. It can serve as a good scaling law handbook for DiLoCo practioners.

* It showed DiLoCo can reach similar PPL as DDP with a much less wall-clock time, due to less communication and larger batchsize

Weakness:

* The work is a re-study of well-known scaling laws like Chinchilla/OpenAI's work, but focused on DiLoCo. The innovation is limited and significance may be limited to DiLoCo practioners

* The wall-clock time saving is only sigfinicant in networks with low bandwidth, e.g., a-b in Figure 4. For network with modern BW, the gain seems mainly from larger batchsize due to outer optimizer, not the sparse synchronization.

---

> ### Author Rebuttal · Authors · 2025-07-29
>
> We would like to correct some misconceptions in the review (especially regarding prior scaling laws). Also, we aren’t sure concretely what the reviewer is suggesting could be improved to warrant acceptance, and would appreciate clarification.
>
> **The work is a re-study of well-known scaling laws like Chinchilla/OpenAI's work, but focused on DiLoCo.**
>
> We respectfully disagree. We do not re-derive the scaling laws of [1, 2]. Rather, we see how the performance of DiLoCo scales with respect to model size $N$ and number of model replicas $M$, simultaneously. We compare it to data-parallel training in the process (and do scaling analysis there), but this is because we want to make sure our results are as accurate as possible. The new content is all in the DiLoCo-specific scaling laws, which are not in [1, 2] whatsoever.
>
> Most of our work and insights are simply not possible to analyze in data-parallel settings. Things like how DiLoCo changes critical batch size, how it affects overtraining budgets, and how hyperparameters should scale across $M$ are simply not axes that are pertinent to data-parallel training. We use some of the same ideas as in [1, 2] (such as fitting power laws to empirical data), but there is a huge body of work on scaling laws that builds on top of the scaling laws of [1, 2] in analogous ways. Moreover, we use unique methodology, like the joint fitting of model size $N$ and model replicas $M$ jointly.
>
> **The innovation is limited and significance may be limited to DiLoCo practitioners.**
>
> Our work is primarily concerned with LLM training under communication constraints. This is a growing area of work, and our focus on DiLoCo within that area is intentional. DiLoCo is demonstrably more effective at reducing communication costs than other LLM training methods [1]. DiLoCo and its variants are already being employed to train large LLMs in a globally distributed manner, both in papers and in actual model deployments (see [2, 3, 4]). In short, DiLoCo is an empirically successful method that already has a large amount of work using it at scale and developing new variants (see [6, 7, 8, 9], as well as others). As such, we believe our work is entirely appropriate for this conference.
>
> More broadly, NeurIPS is a very large conference. There is plenty of research published at NeurIPS which is primarily important to the sub-community that works on it. For example, research on variational inference methods is primarily of interest to the variational inference community. That’s not a limitation of research in that area, it’s just because researchers specialize and have distinct areas of focus.
>
> **The wall-clock time saving is only significant in networks with low bandwidth, e.g., a-b in Figure 4. For network with modern BW, the gain seems mainly from larger batchsize due to outer optimizer, not the sparse synchronization.**
>
> We’re confused as to why this finding, which we believe is an exciting insight in our work, is listed as a weakness. The fact that DiLoCo helps speed up training even over high bandwidth networks is (1) a novel insight and (2) has important implications for LLM training.
>
> We also disagree that LLM training should only consider high bandwidth settings. The high bandwidth setting in Figure 4 is not the type of bandwidth that connects data centers in different parts of the world. Figure 4a is much closer to what you would see in such settings. In fact in [4] the bandwidth available was actually lower than our “low-bandwidth” setting. In short: the bandwidth available across countries is not high enough to train large LLMs purely via data-parallel training, which is why DiLoCo is so promising in realistic LLM training across datacenters.
>
> **At line 185 the authors mentioned an counter-intuitive observation: "outer learning rate increases with H". Do you have any potential explanation for it?**
>
> This is still an open question. One potential explanation is that when you increase H, you decrease the number of outer optimization steps that occur throughout training. To keep the effective total progress the same, you would then need to compensate by increasing the outer learning rate. Also, by increasing H the outer gradients may effectively undergo variance reduction, as they are now aggregations of more training steps, and less variance enables larger learning rates.
>
> ### References
>
> [1] Kaplan et al., “Scaling laws for neural language models,” 2020.
>
> [2] Hoffman et al., “Training Compute-Optimal Large Language Models,” 2021.
>
> [3] Douillard et al., “DiLoCo: Distributed Low-Communication Training of Language Models,” 2023.
>
> [4] Jaghouar et al., “OpenDiLoCo: An Open-Source Framework for Globally Distributed Low-Communication Training,” 2024.
>
> [5] Jaghouar et al., “INTELLECT-1 Technical Report,” 2024.
>
> [6] Sani et al., “The Future of Large Language Model Pre-training is Federated,” 2024.
>
> [7] Kim et al., “HALoS: Hierarchical Asynchronous Local SGD over Slow Networks for Geo-Distributed Large Language Model Training,” 2025.
>
> [8] Therien et al., “MuLoCo: Muon is a practical inner optimizer for DiLoCo,” 2025.
>
> [9] Beton et al., “Improving the Efficiency of Distributed Training using Sparse Parameter Averaging,” 2025.

---

> > ### Comment · Reviewer_6yd8 · 2025-08-02
> >
> > `We’re confused as to why this finding, which we believe is an exciting insight in our work...`
> > My point is the main contribution of diloco per my understand is sparse synchronization yet this finding basically suggests with high bw network there is no need/benefit for sparse synchronization. So this is a weakness from ablation study point of view, though I agree the outer optimizer itself shows a potential in speeding up DDP.
> >
> > And thanks for the detailed reply I am happy to raise my score to 4.

---

### Official Review · Reviewer_Px8m · 2025-07-01

**Clarity:** 3
**Significance:** 1
**Originality:** 2
**Rating:** 3
**Confidence:** 4

**Summary:**

The paper investigate how DiLoCo, a periodic synchronization variant of Local SGD/FedAvg scales when training LLMs under a fixed compute budget. The author adopt the scaling law and conduct empirical studies for evaluation. Key empirical findings are that DiLoCo achieves lower loss than data-parallel training, supports much larger optimal batch sizes, and therefor yields shorter idealized wall-clocak times een on high bandwidth clusters.

Even though the both LLM and scaling laws are timely topic, I think this paper has marginal contribution given the fact that it only studied one particular method, and the fact author adopt the well-established scaling law formulation.

**Questions:**

- Generality of laws: do the hyper-parameters exponents change for MoE or multilingual corpora?

**Ethical Concerns:**

["NO or VERY MINOR ethics concerns only"]

**Final Justification:**

I have carefully reviewed the authors' responses and discussions with other reviewers and AC.

**Limitations:**

Please see above.

**Paper Formatting Concerns:**

No particular concerns.

**Quality:**

2

**Strengths And Weaknesses:**

**Strengths**
- through empirical sweeps

**Weakness**
- the paper only studies one method (DiLoCo), without compare the scalaing laws of other method that are also mentioned in the introduction: FL and data-parallel training. It would be better to have clearly study the difference among them. Also, there are other communication-efficient methods that are not covered in the paper. However, this kind of study suits more of a benchmark paper instead of main track paper in my opinion.

---

> ### Author Rebuttal · Authors · 2025-07-29
>
> We would like to correct some misconceptions in the review (especially regarding data-parallel training). Also, we aren’t sure concretely what the reviewer is suggesting could be improved to warrant acceptance, and would appreciate clarification.
>
> **The paper only studies one method (DiLoCo), without compare the scaling laws of other method that are also mentioned in the introduction: FL and data-parallel training.**
>
> We respectfully disagree with the reviewer's assertion. We did compare DiLoCo with data-parallel training. We do this repeatedly, even re-deriving scaling laws for data-parallel training in our own setting to make sure that our comparisons are as fair as possible.
>
> We’re also not sure what the reviewer is asking for with respect to federated learning (FL). This is a very broad area, not just one algorithm, that could encompass a huge range of things. We mention FL in our introduction because algorithmically, the two areas often use similar optimization algorithms. But most FL settings require things like private heterogeneous client data and privacy safeguarding mechanisms (such as differential privacy), making them very different from LLM pre-training tasks, and not comparable. Most realistic FL is also restricted to very small models due to the need to keep model training on-device.
>
> If the reviewer is asking for us to compare DiLoCo to FedOpt, then would like to note that DiLoCo is FedOpt when the inner optimizer is AdamW and the outer optimizer is Nesterov, which [1] already showed convincingly is the best parameterization of FedOpt when training LLMs.
>
> **Also, there are other communication-efficient methods that are not covered.**
>
> Our focus on DiLoCo is intentional. It is demonstrably more effective at reducing communication costs than other LLM training methods [1]. DiLoCo and its variants are also already being employed to train large LLMs in a globally distributed manner, both in papers and in actual model deployments (see [2, 3, 4, 5]). DiLoCo is being increasingly used as a starting point for new methods in communication-efficient LLM training [6, 7, 8]. Finally, unlike methods such as compression which do not fundamentally alter the training dynamics, the reason DiLoCo is singled out is that it does actually change the underlying algorithmic dynamics. While scaling analyses of things like compression can be done, they rely in large part on their algorithmic similarity to uncompressed training.
>
> **This kind of study suits more of a benchmark paper instead of main track paper.**
>
> We respectfully disagree that scaling law analyses are only suited for benchmark papers. A brief skim of the NeurIPS 2024 papers reveals at least 10 accepted papers working on topics related to scaling laws. Moreover, our work is not simply a benchmarking of methods. It includes novel scaling law methodology (e.g. jointly scaling based on model size and number of model replicas), novel algorithmic insights (e.g. the fact that DiLoCo with a single model replica is a sensible and performant method), and novel axes of comparison (e.g. a focus on horizontal scalability, batch size, and influence of hyperparameters across values of $M$).
>
> **Do the hyper-parameters exponents change for MoE or multilingual corpora?**
>
> We do not know yet whether the scaling laws in our work change when training MoE models or on multilingual data. However, it is computationally infeasible to study all of these in the same paper. These took years and many separate studies to nail down for data parallel training (e.g. scaling laws for MoEs were published only after the Chinchilla scaling laws). Performing scaling law analyses on all of these axes simultaneously is just not possible in a single conference paper. While these are great follow-up studies, they are things that are made easier by our work, not something we believe can be studied all at once.
>
> ### References
>
> [1] Douillard et al., “DiLoCo: Distributed Low-Communication Training of Language Models,” 2023.
>
> [2] Jaghouar et al., “OpenDiLoCo: An Open-Source Framework for Globally Distributed Low-Communication Training,” 2024.
>
> [3] Jaghouar et al., “INTELLECT-1 Technical Report,” 2024.
>
> [4] Sani et al., “The Future of Large Language Model Pre-training is Federated,” 2024.
>
> [5] Kim et al., “HALoS: Hierarchical Asynchronous Local SGD over Slow Networks for Geo-Distributed Large Language Model Training,” 2025.
>
> [6] Therien et al., “MuLoCo: Muon is a practical inner optimizer for DiLoCo,” 2025.
>
> [7] Beton et al., “Improving the Efficiency of Distributed Training using Sparse Parameter Averaging,” 2025.
>
> [8] Qi et al., “DiLoCoX: A Low-Communication Large-Scale Training Framework for Decentralized Cluster,” 2025.

---

> > ### Comment · Reviewer_Px8m · 2025-08-04
> >
> > Many thanks for the authors' rebuttal. As authors mentioned in the first paragraph of the introduction and in the rebuttal, DiLoCo is essentially one particular method of FL. I do agree that some of the FL settings, like FL with DP is not comparable.  However, DiLoCo is not the only way of using FL for LLM pretraining, and there are also other methods that using FL for LLM pretraining [1], which claim to outperform DiLoCo. I still think the study focus on one particular method is too narrow for Neurips.
> >
> > [1] Photon: Federated LLM Pre-Training (https://mlsys.org/virtual/2025/poster/3280)

---

> > > ### Author Response · Authors · 2025-08-04
> > > **Photon does use DiLoCo**
> > >
> > > We want to clarify that the Photon algorithm is DiLoCo (parameterized in exactly the way that we do in the paper) plus a heuristic for setting the batch size (see Algorithm 1 in [1]) and adjusting learning rates.
> > >
> > > We will note that because we do such in-depth tuning and scaling laws of the learning rates and batch size for DiLoCo, this means that we subsume the benefits of Photon in all of our settings. In other words, Photon essentially becomes one operating point in the data we collected to derive scaling laws.
> > >
> > > Again, we would appreciate clarification on what the reviewer believes could warrant acceptance of the paper. We believe that we have addressed all the reviewers' points (we *do* compare to data-parallel, and have recovered Photon through in-depth tuning). We again wish to emphasize that DiLoCo is the building block for most recent work on communication-efficient LLM training, including in the case of [1] (ie. Photon).
> > >
> > > [1] Sani et al., "Photon: Federated LLM Pre-Training," 2024.

---

> > > > ### Comment · Reviewer_Px8m · 2025-08-04
> > > >
> > > > Thanks for the response. I've increase my score to 3.

---

### Official Review · Reviewer_Web4 · 2025-07-03

**Clarity:** 4
**Significance:** 2
**Originality:** 2
**Rating:** 5
**Confidence:** 3

**Summary:**

This paper focuses on the DiLoCo (Distributed Low-Communication) approach for scaling LLM training in a highly distributed setting. This is an important topic due to desire to scale to increasingly large models while harnessing available compute that may exist in different data centers connected with limited bandwidth network connections.

In particular, the goal of this paper is to establish scaling laws for DiLoCo, considering the Chinchilla-optimal number of tokens for a particular model size.

**Questions:**

Q: To what extent do the results reflect more realisitic real-world communication overhead, especially when attempting to train across machines in different data centers?

**Ethical Concerns:**

["NO or VERY MINOR ethics concerns only"]

**Final Justification:**

The work still does not appear particularly creative, but in their response, the authors addressed some of the concerns I had about the utility of this work for potential readers.

**Limitations:**

Yes.

**Quality:**

3

**Strengths And Weaknesses:**

## Strengths

The paper systematically assesses the loss on a held-out set after training, considering HellaSwag, Piqa, and Arc-Easy as zero-shot downstream evaluation tasks.

The loss is varied over model sizes up to 2.4B and over relevant hyperparameters in DiLoCo, including inner and outer DiLoCo learning rate, number of DiLoCo replicas, and batch size.

The derived scaling laws were tested empirically for training 4B and 10B models.

The paper confirms some of the advantages of DiLoCo.

## Weaknesses

Unclear to what extent the scaling law will benefit genuine frontier LLM developers, given some of the simplifications made in this study.

Scaling laws have been derived for a number of architectures and training setups, so the main contribution here is to do this for DiLoCo. In this sense, the study is not particularly creative or exciting. Still, such work can be useful.

---

> ### Author Rebuttal · Authors · 2025-07-29
>
> While we appreciate the positive feedback, we are uncertain what concrete changes the reviewer believes would improve the paper, and would appreciate clarification.
>
> **Unclear to what extent the scaling law will benefit genuine frontier LLM developers, given some of the simplifications made in this study.**
>
> We do not know what simplifications the reviewer is referring to. Additionally, we wish to note that “benefitting frontier LLM developers” is not the acceptance criteria for work at NeurIPS.
>
> That being said, we believe the evidence already shows that our work would be useful to such an audience. For example, [1, 2] constitute open-sourced LLMs trained via DiLoCo already released to the public. If someone wants to scale to larger models, we believe that our work helps them identify how to scale with DiLoCo. Similarly, LLM-focused training libraries like `torchft` and `torchtitan` are already collaborating on how to integrate DiLoCo in order to make LLM training more fault tolerant and scalable from a systems perspective [4].
>
> **To what extent do the results reflect more realistic real-world communication overhead, especially when attempting to train across machines in different data centers?**
>
> We believe our results are extremely useful across a variety of real-world systems. We give 4 specific reasons.
>
> 1. Most of our results are completely independent of the actual bandwidth connection across data centers. The fact that we know how to set hyperparameters algorithmically at larger model sizes $N$ and number of datacenters $M$ is helpful for researchers and practitioners writ large.
> 2. The simulated wallclock times we give have 3 archetypes, as we discuss in the paper: (1) high-bandwidth (400 gigabits / s), (2) medium-bandwidth (100 gigabits / s), (3), low-bandwidth (10 gigabits / s). In fact, we show that DiLoCo is faster across all three, due to increasing the critical batch size. Thus, a huge range of realistic settings are benefitted by DiLoCo.
> 3. Our low-bandwidth is actually an overestimate of the bandwidth used in the DiLoCo training in [1] (see Figure 8, which says the bandwidth they had ranged between 117 and 935 megabits / s).
> 4. As discussed in [3], DiLoCo can significantly improve fault tolerance by enabling greater asynchronicity. Moreover, LLM development libraries like `torchtitan` and `torchft` are already investigating the use of DiLoCo to improve fault tolerance in realistic training settings [4]. Thus, there is a huge amount of utility in showing how DiLoCo scales to large models, where fault tolerance is more important.
>
> ### References
>
> [1] Jaghouar et al., “OpenDiLoCo: An Open-Source Framework for Globally Distributed Low-Communication Training,” 2024.
>
> [2] Jaghouar et al., “INTELLECT-1 Technical Report,” 2024.
>
> [3] Liu et al., “Asynchronous Local-SGD Training for Language Modeling,” 2024.
>
> [4] Rice & Huang, “Fault Tolerant Llama: training with 2000 synthetic failures every ~15 seconds and no checkpoints on Crusoe L40S,” PyTorch blog, 2025.

---

> > ### Author Response · Authors · 2025-08-08
> > **Thoughts on author feedback?**
> >
> > Hello Reviewer Web4. We wanted to see if you had any thoughts on the author feedback above - this is the last day for discussion, and we believe we have addressed all the comments in your review.

---

> > ### Comment · Reviewer_Web4 · 2025-08-08
> > **Thanks**
> >
> > The response addresses some of the concerns that I had about the real-world utility of the findings of this work.
> >
> > I have raised my score.

---

### Official Review · Reviewer_am9z · 2025-07-03

**Clarity:** 2
**Significance:** 3
**Originality:** 4
**Rating:** 5
**Confidence:** 4

**Summary:**

This paper investigates the behavioral patterns of the DiLoCo optimization algorithm [1] with respect to its hyperparameters. The authors analyze how DiLoCo scales compared to Distributed Data Parallel (DDP) across a range of settings. Notably, they observe that the performance gap between DiLoCo and DDP narrows for higher levels of DiLoCo distributed settings as model size increases. Additionally, even when the number of unique model replicas is one, DiLoCo's low-synchronization regime still yields benefits over DDP. Overall, the paper provides a detailed empirical characterization of DiLoCo's performance under varying configurations, highlighting scenarios in which it outperforms/underperforms standard DDP

**Questions:**

Q1.
> “Moreover, our results actually show a potentially counter-intuitive phenomenon: The optimal outer learning rate increases with $H$, even though the outer gradients increase in size as $H$ increases.”

Could the authors clarify why they believe this occurs? Is it perhaps due to a need to ‘overcompensate’ for parameter drift at higher $H$? If so, do the updates also exhibit higher variance?

Q2. Initially, it was unclear why DiLoCo receives a batch size of $B/M$, while DDP receives $B$. According to [1], DiLoCo would shard the $B$ tokens across $M$ workers, just as DDP would distribute them across its workers. The primary distinction seems to be that DiLoCo synchronizes every $H$>1 steps, while DDP synchronizes every step ($H=1)$.
From reading the appendix, I gather that the total number of workers $R$ was fixed for both DDP and DiLoCo. For DiLoCo, these were divided into $M$ clusters, each internally synchronized via DDP. Thus, $M$ does not represent the number of replicas in the traditional sense; rather, it denotes the number of independently synchronized clusters (or the *unique replicas* among the R workers when $H>1$). Under this interpretation, $M=1$ and $H=1$ would approximate standard DDP.
If this is correct, it would be helpful to clarify the definition of $M$ more explicitly in the main text. If not, I would appreciate further clarification, as the current explanation is somewhat confusing.

Q3.
> and evenly partition it at the sequence level across the $M$ DiLoCo replicas.
Why is the partitioning across the sequence level and not batch level for DiLoCo? For pertaining, I can think of cases where this may not be an issue. Maybe I am missing something here, but isn't it possible that this leads to DiLoCo seeing shorter sequences compared to DDP on average?

Q4. It would be valuable to study whether a relationship exists between synchronization cadence and global batch size with respect to evaluation loss. For example, Figure 3 (DDP vs. DiLoCo at large batch sizes) seems to implicitly suggest that a higher $H$ may be preferable at higher global batch sizes. Do the authors observe consistent trends in this regard?

Q5. Similarly, in Figure 15, it's not immediately clear why $H=1$ performs the worst. Is this because the optimizer essentially reduces to solely SGD + Nesterov?

Q6. Are Figures 5 and 6b conveying the same information? If not, it would be helpful to distinguish their respective roles more clearly.


[1] https://arxiv.org/abs/2311.08105

**Ethical Concerns:**

["NO or VERY MINOR ethics concerns only"]

**Final Justification:**

I’m increasing my score to 5: my original concerns have been addressed, but I think broader impact may remain somewhat speculative, as pointed out by fellow reviewers.

**Limitations:**

Yes, the authors discuss limitations.

**Quality:**

2

**Strengths And Weaknesses:**

**Strengths**

S1. Overall, the paper helps build a solid empirical understanding of the behavior of DiLoCo with respect to its key parameters. It provides useful insights into various aspects of its performance, clarifying how one might expect it to behave in practice compared to standard DDP.

S2. The observation that DiLoCo outperforms DDP at $M=1$ is particularly interesting. This suggests that a combination of distinct local and global optimizers may offer benefits even in low-synchronization regimes.

**Weaknesses**

W1. While I acknowledge that the authors explicitly mention the limitation of not validating robustness across different pretraining datasets or model architectures, I believe it remains a critical point. It is important to assess whether claims surrounding some of the observed trends, such as the gains at low $M$, can be expected to generalize across training settings.

W2. Some sections lack clarity or are not well written (see Questions).

W3. Certain results and observations could benefit from more thorough justification and analysis (see Questions).

---

> ### Author Rebuttal · Authors · 2025-07-29
>
> We thank the reviewer for their detailed feedback. We believe that most of the points raised can be directly addressed via minor sentence rewriting. The remaining we have addressed to the best of our ability (and computation) below.
>
> **While the authors explicitly mention the limitation of not validating robustness across different pretraining datasets or model architectures, I believe it remains a critical point. It is important to assess whether claims surrounding some of the observed trends, such as the gains at low M, can be expected to generalize across training settings.**
>
> While we agree that generalizability is important, we are confident in our results for a variety of reasons.
>
> 1. Our architecture and data were chosen explicitly to aid generalizability. We use the exact architecture style from NanoDO [1]. There have been a wide array of scaling work predicated on NanoDO, that we believe showcase the reliability of our results, including [2-9]. The training setup has generated many insights into transformer pre-training, enhancing our confidence in the results. Our primary training dataset, C4, has also been widely used across a huge variety of language model training settings.
> 2. The utility of $M=1$ is intuitive when considering the empirical success of the lookahead optimizer [10]. We present a tuned version that incorporates momentum. Note that if our tuning is done well, it should do at least as well as the lookahead optimizer (which it does) and in fact does better. Given the success of SGDM over SGD, this is in retrospect, not surprising, but not something that has been shown in the literature before.
> 3. We saw qualitatively similar results on both the C4 training dataset and the Dolma dataset (which we used for the overtraining ablations).
> 4. The amount of compute needed to duplicate our results on a different set of model architectures would be quite intensive. The amount of experimental work that went into this was already quite significant, and to add more to a single conference paper would be a lot.
> 5. The relative success of DiLoCo has been seen in many works. While ours adds a new scaling dimension, the results for a fixed model size mirror those in other works on DiLoCo [11, 14, 15].
> 6. The formative works on scaling laws [12, 13] used similar architectures and datasets, and have been extremely generalizable.
>
> In short, we have (short of doubling our computational expenditure) done our best to ensure generalizability of results.
>
> **Q1: Could the authors clarify why optimal outer learning rate increases with H?**
>
> This is an open question. One potential explanation is that when you increase $H$, you decrease the number of outer optimization steps that occur throughout training. To keep the effective total progress the same, you would then need to compensate by increasing the outer learning rate. Additionally, by increasing $H$ the outer gradients may effectively undergo variance reduction, as they are now aggregations of more training steps, enabling larger learning rates.
>
> **Q2: What is the distinction between DiLoCo when H = M = 1 and DDP?**
>
> We discuss this in L96-103 and L111-113 in the paper. The only difference when $M = H = 1$ is the use of an outer optimization step that incorporates momentum. If OuterOpt in Algorithm 1 is SGD with learning rate 1 and no momentum, we recover DDP when $H = M = 1$. This is also noted in [11]. We will mention this fact explicitly.
>
> **Q3: Why is the partitioning across the sequence level and not batch level for DiLoCo? For pertaining, I can think of cases where this may not be an issue. Maybe I am missing something here, but isn't it possible that this leads to DiLoCo seeing shorter sequences compared to DDP on average?**
>
> By “partitioning at the sequence level” we mean that if a batch consists of $B$ sequences of tokens, we give each replica $B/M$ of those sequences. We said “sequence level” to indicate that we are preserving entire sequences, so our partitioning was not at the “token level”. In short, we do exactly the same partitioning that DDP does. We will make sure to clarify this.
>
> **Q4: It would be valuable to study whether a relationship exists between synchronization cadence and global batch size with respect to evaluation loss.**
>
> We agree that this is an interesting study. Unfortunately, it is not possible to tune synchronization cadence $H$ and global batch size $B$ without also tuning inner learning rate $\gamma$ and outer learning rate $\eta$. Simultaneously tuning $H$, $B$, $\gamma$, and $\eta$ across model sizes is even more computationally intensive than the experiments in our paper (due to the extra parameter $H$). In the case of our studies on $H$ in Section 5, we did not re-tune $\gamma$ and $B$ when varying $H$, as otherwise it would have been too expensive to run.
>
> **Q5: In Figure 15, it's not immediately clear why H=1 performs the worst. Is this because the optimizer essentially reduces to solely SGD + Nesterov?**
>
> We believe that the answer is analogous to why the optimal outer learning rate increases with $H$. When $H=1$, the outer gradients are likely quite noisy (as noisy as the inner gradients), and so the momentum (on top of the inner AdamW optimization already happening) isn’t useful. By contrast, when $H > 1$ you get a form of variance reduction that we conjecture aids optimization.
>
> **Q6. Are Figures 5 and 6b conveying the same information? If not, it would be helpful to distinguish their respective roles more clearly.**
>
> We discuss this in L177-188. We state explicitly “We first study whether the observation in Section 4, that $\eta$ should be tuned independently of $N$, holds for other values of $H$”. We are happy to explicitly reference Figure 5 in this line, instead of implicitly. The key addition in Figure 6b is varying $H$.
>
> ### References
> [1] Liu et al., “NanoDO: A minimal Transformer decoder-only language model implementation in JAX”, 2023.
>
> [2] Wortsman et al., “Small-scale proxies for large-scale transformer training instabilities,” 2023.
>
> [3] Everett et al., “Scaling Exponents Across Parameterizations and Optimizers,” 2024.
>
> [4] Lechao Xiao, “Rethinking Conventional Wisdom in Machine Learning: From Generalization to Scaling,” 2024.
>
> [5] Schug et al., “Attention as a Hypernetwork,” 2024.
>
> [6] McKenna et al., “Scaling Laws for Differentially Private Language Models,” 2025.
>
> [7] Marek et al., “Small Batch Size Training for Language Models: When Vanilla SGD Works, and Why Gradient Accumulation Is Wasteful,” 2025.
>
> [8] Gyorgy et al., “Beyond Statistical Learning: Exact Learning Is Essential for General Intelligence,” 2025.
>
> [9] Roulet et al., “Loss Functions and Operators Generated by f-Divergences,” 2025.
>
> [10] Zhang et al., “Lookahead Optimizer: k steps forward, 1 step back,” 2019.
>
> [11] Douillard et al, “DiLoCo: Distributed Low-Communication Training of Language Models,” 2023.
>
> [12] Kaplan et al., “Scaling laws for neural language models,” 2020.
>
> [13] Hoffman et al., “Training Compute-Optimal Large Language Models,” 2021.
>
> [14] Jaghouar et al., “OpenDiLoCo: An Open-Source Framework for Globally Distributed Low-Communication Training,” 2024.
>
> [15] Jaghouar et al., “INTELLECT-1 Technical Report,” 2024.

---

> > ### Comment · Reviewer_am9z · 2025-08-04
> >
> > While I agree with fellow reviewers that the scope of this work may currently feel limited to DiLoCo practitioners, I also recognize that DiLoCo, though not widely adopted yet, has the potential to gain broader traction. This paper, by effectively demonstrating DiLoCo's scalability, could be the catalyst which shifts the perception.
> >
> > So I’m increasing my score to 5: my original concerns have been addressed, but I think broader impact may remain somewhat speculative.

---

### Decision · Program_Chairs · 2025-09-17

**Decision:**

Accept (spotlight)

**Comment:**

The paper focuses on DiLoCo (Distributed Low-Communication), a periodic-synchronization optimization algorithm for scaling LLM training, and aims to fill a critical gap: systematic empirical characterization of its hyperparameter behavior and scaling laws relative to standard Distributed Data Parallel (DDP).

Reviewer Px8M and 6yd8 recognized a few weaknesses, but in short sentences and with low confidence. Reviewer am9z and Web4 give more positive feedback.

Post-rebuttal, the authors effectively addressed all critical reviewer concerns by clarifying the scope, improving readability, and justifying their focus on DiLoCo’s depth over breadth. While the paper is not theoretically groundbreaking, its empirical rigor and actionable guidance for LLM training at scale make it a valuable addition to the conference. Considering the overall positive scores, I recommend acceptance.